# Mathematical Models for FDG Kinetics in Cancer: A Review

**DOI:** 10.3390/metabo11080519

**Published:** 2021-08-06

**Authors:** Sara Sommariva, Giacomo Caviglia, Gianmario Sambuceti, Michele Piana

**Affiliations:** 1Life Science Computational Laboratory (LISCOMP), Largo Rosanna Benzi 10, 16132 Genova, Italy; sommariva@dima.unige.it; 2Dipartimento di Matematica, Università di Genova, Via Dodecaneso 35, 16146 Genova, Italy; caviglia@dima.unige.it; 3CNR-SPIN, Corso Perrone 24, 16152 Genova, Italy; 4Dipartimento di Scienze della Salute, Università di Genova, Largo Rosanna Benzi 10, 16132 Genova, Italy; sambuceti@unige.it; 5Ospedale Policlinico San Martino IRCCS, Largo Rosanna Benzi 10, 16132 Genova, Italy

**Keywords:** Positron Emission Tomography (PET), 2-deoxy-2-[18F]fluoro-D-glucose (FDG), tracer kinetics, compartmental analysis

## Abstract

Compartmental analysis is the mathematical framework for the modelling of tracer kinetics in dynamical Positron Emission Tomography. This paper provides a review of how compartmental models are constructed and numerically optimized. Specific focus is given on the identifiability and sensitivity issues and on the impact of complex physiological conditions on the mathematical properties of the models.

## 1. Introduction

Glucose plays a crucial and for a large part unexplained role in cancer metabolism [1]. Indeed, a common feature of tumor pathological metabolism is an increased glucose uptake, together with its fermentation to lactate, even under aerobic conditions [2]. This behavior is known as Warburg effect [3], and the revealing of its mechanism and function is one of the most intriguing open issues of cancer biochemistry in nearly one hundred years.

2-deoxy-2-[18F]fluoro-D-glucose (FDG) [4] is a glucose analog that is systematically utilized as a radioactive tracer in nuclear medicine. In fact, once injected in a living organism, FDG is carried to tissues by blood, is diffused in tissues, is transported into cells by the same transporters (GLUTs) as glucose, and is eventually trapped into cells after phosphorylation by hexokinase (HK). Further, Warburg effects increase the FDG consumption by cancer cells, which makes this tracer useful for cancer detection and staging, and for the assessment of clinical therapies.

FDG Positron Emission Tomography (FDG-PET) [5] is a functional imaging modality that utilizes FDG as a tracer in order to quantitative assess FDG metabolism in tumors (but other pathologies are systematically investigated, as well, by means of this imaging technique). FDG-PET measures the radiation emitted by the tracer injected in the organism, and these measurements encode, in a very indirect way, two kinds of information: the localization of FDG accumulation in the body and the rate with which FDG changes its metabolic status along time. In order to decode such sophisticated information, two inverse problems must be solved:Image reconstruction inverse problem [6]: to reconstruct the spatio-temporal distribution of FDG inside the tissue by solving the integral equation that connects the FDG densito to the measured radiation by means of the Radon transform.Compartmental inverse problem [7]: to model the tracer kinetics by solving the non-linear time-dependent equation that connects the tracer coefficients to the reconstructed FDG concentration.

The focus of the present review is on the compartmental inverse problem. In this framework, on the one hand, the recorded tissue activity corresponds to the superposition of the tracer signal in the blood, in the interstitial tissue, and within the cell. On the other hand, the unknowns are the parameters associated to the kinetics of the tracer, which mimics the kinetics of glucose. From a mathematical perspective, compartmental models rely on the law of concentration conservation between functionally homogeneous conditions; this leads to a Cauchy problem whose number of ordinary differential equations corresponds to the number of compartments. In this Cauchy problem, the constant coefficients provide the rate of tracer flow between compartments and, since they are able to describe the action of the enzymes metabolizing the tracer, they represent the unknowns of the compartmental problem. At a more specific level, typical technical assumptions are that:Only one compartment is allowed to exchange tracer with the environment.The input function, i.e., the tracer concentration introduced into the tissue by the blood, is known by means of either experimental measurements or mathematical modeling.The overall tracer concentration associated to the organ of interest (typically, the tumor) is known as a function of time.Both the linearity of fluxes between compartments and vanishing initial conditions hold.The kinetic coefficients are constant and homogeneous in the tissue.

Within this framework, the present review will consider just basic schemes with a limited number of variables. For them, we will provide a description of the input data (Section 2) and of how the compartmental models can be constructed (Section 3). A specific focus will be given to standard graphical approaches and to the connection between them and the more general compartmental analysis (Section 4). Further, we will discuss the consequences of the intrinsic ill-posedness of compartmental problems, including their lack of uniqueness and of sensitivity (Section 5). We will also illustrate some specific compartmental models that are related to as much specific physiological conditions (Section 6). And we will briefly review some of the optimization methods that have been formulated and implemented in order to reduce such models (Section 7). Our conclusions will be offered in Section 8.

## 2. The Experimental Data in the Compartmental Game

Solid tumors present an increase of the glucose consumption that occurs even under aerobic conditions [8].This mechanism is known as the Warburg effect, and its origin is still unknown, although several studies have shown a significant correlation between glucose consumption and tumor aggressiveness. From an experimental viewpoint, it is very difficult to directly measure the flux of glucose molecules through biological tissues populated by cancer cells; however, the peculiar kinetic features of a glucose analog, the 2-[18F]-2deoxy-D-glucose (FDG), allow a reliable estimate of such parameter. Indeed, FDG is transported through cell membranes by the same GLUT transporters as glucose, and it is trapped in the cytosol by phosphorylation catalyzed by the same hexokinases. However, differently from glucose-6-phosphate (G6P), FDG6P is a false substrate for downstream enzymes channeling G6P to glycolysis or the pentose-phosphate pathway. Thus, FDG6P accumulates in cells and tissues, and its amount is considered an accurate marker of their overall glucose consumption [9,10,11]. Accordingly, the measured tracer content may be employed in non invasive cancer detection and staging, and in the assessment of drug treatments.

In vivo, cancer FDG uptake depends on the glucose level in the blood [12,13], on the possible administered drugs [14], and on the amount of the available tracer, which in turn is related to the diffusion processes occurring in the whole body after injection. Further, tracer concentration in the blood is a time-dependent parameter, as a consequence of several factors, such as absorption by the brain [15], urinary elimination [14], accumulation in liver [16], and the different accumulation rates of the various tissues [17].

In Positron Emission Tomography (PET), the tracer is injected into the subject via intravenous administration, and gamma-ray detectors measure the radiation emitted in vivo by the target tissue. This device is calibrated in such a way that imaging procedures allow the reconstruction of the tracer distribution in the body. The reconstructed activity concentration may depend on both a specific region of interest at a chosen time, and on the time course in a given time interval, where the time variable *t* is measured in minutes from the time of the tracer injection.

In this section we recall a few essential features of the sources of data providing the input for kinetic models of tracer dynamics. We consider first the standardized uptake value (SUV), which approximates the tracer metabolic assessment by using a single time frame (static imaging); next we consider (dynamic) estimates of the time course of tracer concentration in blood (input function) and tracer concentration in the target tissue, which are both obtained from a time series of images.

### 2.1. Standardized Uptake Value

Perhaps, the standardized uptake value (SUV) is the simplest parameter which is used to quantify tracer accumulation from reconstructed PET images. First, the concentration of tracer emitters in a region of interest (ROI) of the target tissue is recovered at a given time, as the solution of an appropriately defined inverse problem. Next, the corresponding normalized radioactivity concentration is estimated by the SUV, which is defined as [18]
$$\mathrm{SUV}=\frac{\mathrm{activity}\;\mathrm{concentration}\;\mathrm{per}\;\mathrm{unit}\;\mathrm{mass}\;\left[\mathrm{Bq}/\mathrm{kg}\right]}{\mathrm{injected}\;\mathrm{activity}\;[\mathrm{Bq}]/\;\mathrm{body}\;\mathrm{mass}\;\left[\mathrm{kg}\right]}.$$

According to this definition, the radioactivity concentration in the ROI is normalized to the radioactivity concentration in the body, which is estimated as the ratio between the injected activity and the patient body mass. There are slightly different definitions available; moreover, SUV measurements are affected by physiological and technological factors [17,18].

Overall, the SUV is an oversimplified index depending on the time interval between injection and observation, location and dimensions of the ROI, and uptake by other tissues [17]. Nevertheless, the localized SUV has been used to stage tumors and to assess response to therapies.

### 2.2. Input Function

Tissues extract the tracer from blood. Indeed, only free FDG is available for tissue uptake, while the radioligand bound to blood cells and metabolites is firmly constrained to remain in blood [19,20]. In the present work, the concentration $C_{b}$ of the tracer available for input to tissues is identified with the measured concentration in arterial blood, which means in particular that the bound tracer is disregarded. Considerations about bound tracer are discussed in Reference [20].

Tracer is delivered to tissues via blood flow, so that the amount of tracer locally extracted by a tissue is highly dependent on the concentration of radioactivity in blood. Thus, the reconstruction of FDG kinetics requires the knowledge of the arterial plasma time-activity concentration curve of the tracer, which in turn provides an estimate of the radioactivity available for uptake. There are several ways for the determination of the time course of concentration: serial arterial sampling, which is independent of PET data acquisition; images of tracer concentration in blood pools [21], such as the left ventricle; a variety of statistical reconstruction methods [22].

In the present work, the arterial plasma activity concentration curve is regarded as given and plays the role of input function (IF). We assume that the IF, as well as any other activity curve, has been corrected for tracer decay.

### 2.3. Activity Concentration of Target Tissue

The target tissue (TT) is the tissue selected for measuring the activity concentration $C_{T}$ [kBq/mL]. Specifically, the tissue response is the time course of $C_{T}$, which is computed by realizing a ROI-based analysis of a dynamical series of reconstructed PET images [9,23]. Of course, these data are corrected for attenuation and possible systematic errors, and the resulting $C_{T}$ depends on several factors like the injected dose, the shape of the IF, the characteristics of the TT, and the pato-physiological conditions of the patient.

We point out that the activity concentration measured by a PET device is the superposition of different signals: the one emitted by tracer molecules in the blood that occupies the ROI volume, the one associated to the tracer in the interstitial volume, and the one emitted by the FDG molecules phosphorylated in the cell. The analysis of these PET data aims at the reconstruction of the detailed kinetics of the tracer. In a sense, the measured signal has to be resolved into the activity pertaining to each source. To this aim, tracer kinetics takes into account the flow of radioactive molecules between the various sources. This is achieved by the application a mathematical models, as described in the next section. Comparison between model predictions and measured data leads to the determination of tracer kinetics, through solution of an inverse problem.

## 3. The Construction of Compartmental Models

### 3.1. Generalities

Compartmental models relate the measured dynamical PET data to either specific metabolic states or tracer chemical compounds, also accounting for their distribution in space. These functional states are known as compartments (or also as sources, or pools). Compartmental analysis relies on the so-called well-mixed assumption, stating that the tracer distribution in each compartment must be considered as homogeneous, and the tracer is instantaneously mixed when it is exchanged between compartments. A discussion of further assumptions will be described in detail in the next subsection.

In the framework of compartmental analysis, compartments have specific functional meanings and are characterized by specific time-dependent tracer concentrations. Therefore, different compartments may occupy the same spatial volume as typically occurs, for instance, in the case of free and phosphorylated FDG molecules. Conversely, a specific chemical compound may be associated to multiple spatially distinguished compartments and, in them, it may be characterized by different concentrations.

A compartmental model is a set of interconnected compartments, whose number depends on the chemical, physiological, and biological properties of the tracer [7,9,24]. In the compartmental framework, a biological system with complex physiological properties is approximated by a limited number of basic functional constituents. In this context, even the blood could be regarded as a compartment; however, in this review, we will assume that the tracer concentration in the blood is known and provided by ad hoc measurements [23].

The concentrations of the various pools are the state variables of the CM, their time dependence being determined by tracer exchange. The tracer flux between compartments, e.g., from the free to the phosphorylated pool, occurs according to mass conservation. Usually, it is assumed that the outgoing flux depends on the concentration of the source.

Assuming that the conservation of the tracer concentration occurs implies that the time rate of the tracer concentration for each compartment is given by the difference between the amount of tracer entering the pool and the amount of tracer leaving the same pool, per unit time and unit volume. This assumption, therefore, leads to a system of ordinary differential equations (ODEs), in which the IF providing the tracer supply to the compartments represents the forcing function of the system. We also assume that all the initial concentrations are zero, i.e., that there is no tracer available to the tissue at the starting point of the experiment. In the resulting mathematical model, the state variables represent the solutions of the Cauchy problem, which is made of linear ODEs with constant and homogeneous rate coefficients (also named transfer coefficients or microparameters) representing the rate of tracer flux between compartments, i.e., the phosphorylation rate for the FDG molecules [25]. Given the rate coefficients and the initial state, the numerical solution of the Cauchy problem describes the tracer kinetics. However, in compartmental analysis, the rate coefficients are unknown and must be determined in such a way that the resulting estimates are in accordance with the measured overall tissue concentration. Therefore, the compartmental problem is intrinsically a non-linear inverse problem, and the first step for its solution is the proof that these rate coefficients can be uniquely determined from the measured data. This need implies a limitation on the number of microparameters and, in turn, on the number of functional compartments. The result of this approach is a trade-off between an exhaustive accordance to reality, and the need of simplifying the formal description and the corresponding system of equations.

In the course of this section we first recall the basic conditions needed to ensure the applicability of compartmental analysis. The general reference framework has been described by the previous discussion, which also provides some motivation, and, with some more detail, in [26]. Next, we examine a few CMs, which are of rather common use, and hence are regarded as highly significant. Particular attention is devoted to the formulation of the inverse problem equation (IPE), which provides the starting point for the formulation of the inverse problem, whose solution determines the kinetics of the tracer. Finally, we introduce a compact (matrix) form of CMs, we present the formal solution of the direct problem, and we write down the IPE in a general form.

### 3.2. Basic Applicability Conditions

In a typical PET experiment, a fraction of tracer is adsorbed by tissues after injection into blood, while some tracer is lost by tissues and poured back into blood. The previous discussion has indicated that application of compartmental analysis allows a reliable reconstruction of tracer kinetics, but this can be achieved only if a certain number of conditions are satisfied in the course of the experiment [9,10,19,23,27]. The following list describes the most common and relevant requirements.
Tracer is administered in trace amounts. The number of injected molecules is supposed to be sufficiently high so that diffusion may described by application of a continuous model. However, such a number is not so high as to influence physiological processes and molecular interactions. In particular, tracer does not affect glucose metabolism.During an experiment, physiologic conditions are in a steady state which is not affected by measurement devices of tracer concentration. This holds true, in particular, for glucose metabolism.The well-mixing condition holds for each compartment. In practice, this means that equilibrium is reached in a time interval, which is rather short with respect to the time of data acquisition. As a consequence, the spatial homogeneity condition follows, which implies that the tracer concentration in each compartment depends only on time.Transport of tracer molecules and related composites between compartments follows a first order kinetics, which ultimately leads to linear ODEs.Bound tracer in blood is disregarded, and the arterial concentration of tracer available for tissue uptake is regarded as a valuable approximation of capillary concentration.

We have described general assumptions underlying most used compartmental models. More specific aspects of tracer kinetics may be considered in order to generate highly realistic models. For example, a distinction may be introduced between free interstitial tracer and free intracellular tracer; permeabilities of blood vessels and cellular membranes may be considered, as well as dependence of activity on spatial variables. CMs explicitly devoted to the modeling of particular physiologic conditions will be examined in a subsequent section.

### 3.3. Examples of Standard CMs

The following examples are itemized according to growing complexity. We adopt typical notations and conventions of the nuclear medicine framework. To simplify, compartments and corresponding concentrations [kBq/mL] are denoted by capital letters $C_{a}$ and $C_{a}$, respectively, where the low index *a* identifies the specific compartment. Rate constants [min${}^{-1}$] are denoted by $k_{b}$, where the low index refers to the specific function in the set of interconnected compartments. Equal low indexes in different compartment models correspond to the same interpretation; at each step, interpretations already discussed for the previous steps are not repeated. We recall that, unless otherwise specified, the initial state is always considered at vanishing concentrations. In figures, compartments are denoted as boxes; by a slight abuse, the same box notation is adopted for the blood compartment; arrows represent flux of tracer between compartments; superposed indexed letters identify the rate constants.

#### 3.3.1. 1-Compartment Model

The 1-compartment model (1-CM) was proposed in order to quantify blood perfusion [23,27]; recently, it has also been applied to describe tracer exchange during the flow of blood through a capillary [28].

The 1-CM is an oversimplified model where there is only one tissue compartment $C_{f}$ with state variable $C_{f}$, accounting for the overall tracer content. The concentration $C_{b}$ of the IF is given. The differential equation for $C_{f}$ is
(1)$${\dot{C}}_{f}=-k_{2}\;C_{f}+k_{1}\;C_{b}\;,$$
where $k_{1}$ and $k_{2}$ are the rate constants for the incoming and outgoing tracer. In other words, $k_{1}\;C_{b}$ is the rate of the incoming flow of tracer per unit volume, while $k_{2}\;C_{f}$ is the analogous rate of the outgoing flow; thus, the net rate of tracer concentration per unit volume, ${\dot{C}}_{f}$, is the difference $k_{1}C_{b}-k_{2}C_{f}$, consistently with conservation of the tracer mass. Notice explicitly that the plus and minus signs refer systematically to incoming and outgoing flows, respectively, for the compartment considered. The case $k_{2}=0$ corresponds to irreversible uptake, which means that tracer cannot escape from the compartment.

The solution of Equation (Equation 1) (vanishing at t=0) is
(2)$$C_{f}=k_{1}\;\int_{0}^{t}e^{-k_{2}\;\left(t-\tau \right)}\;C_{b}\left(\tau \right)\;d\tau .$$

Therefore, the concentration $C_{f}$ is proportional to $k_{1}$, which is related to the absorption capacity of the tissue. The parameter $1/k_{2}$ is related to the asymptotic equilibrium time. For example, in the case $C_{b}\left(t\right)=\bar{C}$ constant, it is found that $C_{f}=h\;\left(1-e^{-k_{2}\;t}\right)$, where $h=\bar{C}\;k_{1}/k_{2}$ denotes the asymptotic equilibrium value. If $t=1/k_{2}$, it is found that $C_{f}=0.63\;h$, rather close to the asymptotic value.

Consider the measured activity concentration per unit volume, $C_{T}$. This PET measurement is set equal to the weighted sum of the free tracer activity concentration in tissue and a contribution arising from the distributed blood and vessels, of the same concentration $C_{b}$ as that of the whole blood. Thus, we have
(3)$$C_{T}=\left(1-V_{b}\right)\;C_{f}+V_{b}\;C_{b}\;,$$
where the dimensionless parameter $V_{b}$ denotes the volume fraction of tissue occupied by blood. Replacing (Equation 2) into (Equation 3) provides the IPE for the 1-CM for the unknown rate constants $k_{1}$ and $k_{2}$.

#### 3.3.2. 2-Compartment Model

Figure 1 depicts a standard 2-compartment model (2-CM) that has been used to model the intracellular processes of phosphorylation and dephosphorylation of FDG through two compartments, $C_{f}$ and $C_{p}$, accounting for free and phosphorylated tracers, respectively [23,29]. The corresponding linear system of ODEs involves two state variables, namely the concentrations $C_{f}$ and $C_{p}$, and four rate constants, and is written as
(4)$$\begin{array}{c} {\dot{C}}_{f}=-\left(k_{2}+k_{3}\right)\;C_{f}+k_{4}\;C_{p}+k_{1}\;C_{b}, \end{array}$$
(5)$$\begin{array}{c} {\dot{C}}_{p}=k_{3}\;C_{f}-k_{4}\;C_{p}. \end{array}$$

The two equations express the conservation of tracer exchanged between the compartments $C_{f}$ and $C_{p}$. Thus, the outgoing flux $-k_{3}\;C_{f}$ from $C_{f}$ in (Equation 4) corresponds to the incoming flux $k_{3}\;C_{f}$ into $C_{p}$ in (5). Similar remarks hold for the flux $k_{4}\;C_{p}$. The constants $k_{3}$ and $k_{4}$ are related to phosphorylation and dephosphorylation rates, respectively, resulting from the action of the enzymes HK and G6Pase. Again, $k_{1}$ and $k_{2}$ are the rate constants for the incoming and outgoing tracer between the compartment $C_{f}$ and blood.

The case $k_{4}\neq 0$ is usually referred to as the reversible 2-CM, since the tracer is allowed to leave the compartment $C_{p}$. Conversely, the case $k_{4}=0$ corresponds to the irreversible 2-CM, in the sense that dephosphorylation does not occur; hence, the tracer is trapped inside $C_{p}$; as a consequence, the corresponding concentration $C_{p}$ grows in time. In general, a compartmental system is said to be irreversible if it contains at least one irreversible compartment; otherwise, it is called reversible. The irreversible 2-CM was considered first in a seminal paper by Sokoloff et al [29]. A great number of reconstruction of rate constants performed over years have shown that, in general, the value of $k_{4}$ is relatively small, so that the irreversible 2-CM is often regarded as a mathematical model providing a satisfactory approximation of tracer kinetics.

For later convenience, we observe that the solution of the system (Equation 4), (5) in the irreversible case ($k_{4}=0$) takes the form
(6)$$C_{f}=k_{1}\;\int_{0}^{t}e^{-\left(k_{2}+k_{3}\right)\;\left(t-\tau \right)}\;C_{b}\left(\tau \right)\;d\tau$$
(7)$$C_{p}=\frac{k_{1}\;k_{3}}{k_{2}+k_{3}}\;\int_{0}^{t}C_{b}\;d\tau -\frac{k_{3}}{k_{2}+k_{3}}\;C_{f}\;.$$

The solution of the system (Equation 4), (5) in the reversible case ($k_{4}\neq 0$) is given as
(8)$$C_{f}=\frac{k_{1}}{\lambda_{1}-\lambda_{2}}\;\left[\left(k_{4}+\lambda_{1}\right)\;I_{1}-\left(k_{4}+\lambda_{2}\right)\;I_{2}\right]$$
(9)$$C_{p}=\frac{k_{1}\;k_{3}}{\lambda_{1}-\lambda_{2}}\;\left[I_{1}-I_{2}\right].$$
where
(10)$$I_{1}=\int_{0}^{t}e^{\lambda_{1}\;\left(t-\tau \right)}\;C_{b}\left(\tau \right)\;d\tau ,\;I_{2}=\int_{0}^{t}e^{\lambda_{2}\;\left(t-\tau \right)}\;C_{b}\left(\tau \right)\;d\tau$$
with
(11)$$\lambda_{1,2}=\left[-\left(k_{2}+k_{3}+k_{4}\right)\pm \sqrt{\Delta }\right]/2,$$
and
$$\Delta ={\left(k_{2}+k_{3}+k_{4}\right)}^{2}-4\;k_{2}\;k_{4}={\left(k_{2}+k_{3}-k_{4}\right)}^{2}+4\;k_{3}\;k_{4}.$$

In other words, the concentrations are expressed as linear combination of the integrals $I_{1}$ and $I_{2}$, with coefficients depending on the rate constants.

The connection between the mathematical model and the PET data is given by the IPE
(12)$$C_{T}=\left(1-V_{b}\right)\;\left(C_{f}+C_{p}\right)+V_{b}\;C_{b}.$$

The concentration $C_{T}$ in the left side of (Equation 12) refers to the total activity of the target tissue, reconstructed from the analysis of PET images; the contribution $V_{b}\;C_{b}$ in the right side comes from tracer molecules in blood, while $\left(1-V_{b}\right)\;\left(C_{f}+C_{p}\right)$ comes from tracer molecules in tissue cells: here $C_{b}$ is given, while $C_{f}$ and $C_{p}$ are the formal solutions of the system of ODE, expressed in either the irreversible form (Equation 6), (Equation 7), or the reversible form (Equation 8), (Equation 9). Overall, Equation (Equation 12) expresses the fact that the measured tracer concentration, in the ROI considered, results from FDG molecules in blood, and free and phosphorylated FDG in tissue and molecules.

We point out that there are situations where the tracer content of blood may be disregarded, which corresponds to setting $V_{b}=0$. Conversely, if needed, the volume fraction $V_{b}$ may be regarded as an additional unknown parameter.

#### 3.3.3. 3-Compartment Model

Biochemistry explains that the endoplasmic reticulum (ER) [30] plays a crucial role in the activation of G6Pase and that, specifically, the hydrolysis of G6P and FDG6P with the corresponding creation of free glucose and FDG molecules and a phosphate group occurs after the transmembrane protein glucose-6-phosphate transporter (G6PT) transports the phosphorylated forms into ER [31]. After this process, free FDG can be released into the cytosol. This argument and further biochemical, pharmacological, clinical, and genetic data implies the interpretation of ER as a specific functional compartment [32].

Driven by biochemical reactions involving FDG molecules, a 3-compartment model (3-CM) has been developed, which is formed by the following compartments: $C_{f}$, accounting for free tracer in the cytosol and possibly in the interstitial space, $C_{p}$ for phosphorylated FDG in cytosol, and $C_{r}$ for phosphorylated FDG in ER [13,33]. The resulting compartment model is depicted in Figure 2. The standard compartmental 2-CM is recovered from the 3-CM under the assumption that the ER compartment is removed.

3-CM has been applied to the reduction of data coming from in vitro [13] and in vivo [33] experiments. Although we deal with the same set of three compartments, the resulting models and the related approaches differ in various significant aspects. For example, the process of data acquisition in vitro requires the use of a dedicated device (Ligand Tracer) with a properly defined calibration process, while activities replace concentrations as state variables [13,33]. However, the most outstanding result obtained from the mathematical analysis of data via 3-CM, i.e., that tracer is accumulated in ER, holds both in vitro and in vivo, and this has been confirmed by the localization of fluorescent FDG analogues in the case of in vitro experiments [13].

Coherently with the general approach of this review, in this section we restrict our attention to applications of 3-CM to the analysis of in vivo data.

The state variables are the concentrations $C_{f}$, $C_{p}$, and $C_{r}$. The system of ODEs is given by
(13)$$\begin{array}{c} {\dot{C}}_{f}=-\left(k_{2}+k_{3}\right)\;C_{f}+k_{6}\;C_{r}+k_{1}\;C_{b}, \end{array}$$
(14)$$\begin{array}{c} {\dot{C}}_{p}=k_{3}\;C_{f}-k_{5}\;C_{p}, \end{array}$$
(15)$$\begin{array}{c} {\dot{C}}_{r}=k_{5}\;C_{p}-k_{6}\;C_{r}. \end{array}$$

The rate constant $k_{5}$ is related to tracer transport across the membrane of the ER, or the action of the transporter G6PT. The parameter $k_{6}$ is related to dephosphrylation by G6Pase; thus it may be regarded as the correspondent of $k_{4}$ in the 2-CM system. Since dephosphorylation occurs only inside ER, the parameter $k_{4}$, which corresponds to a flux from $C_{p}$ to $C_{f}$, is set equal to 0.

Following the previous definitions, the case $k_{6}\neq 0$ is referred to as the reversible 3-CM, since the tracer is allowed to leave the ER after dephosphorylation. Conversely, the case $k_{6}=0$ corresponds to the irreversible 3-CM: dephosphorylation does not occur, and the tracer cannot leave the ER compartment.

For later reference, we observe that the solution of the irreversible version ($k_{6}=0$) of the system (13)–(15) may be written as
(16)$$C_{f}=k_{1}\;\int_{0}^{t}e^{-\left(k_{2}+k_{3}\right)\;\left(t-\tau \right)}\;C_{b}\left(\tau \right)\;d\tau ,$$
(17)$$C_{p}=k_{3}\;\int_{0}^{t}e^{-k_{5}\;\left(t-\tau \right)}\;C_{f}\left(\tau \right)\;d\tau ,$$
and
(18)$$C_{r}=k_{5}\;\int_{0}^{t}\;C_{p}\left(\tau \right)\;d\tau \;.$$

The connection between the tracer concentration $C_{T}$ estimated over a suitable ROI and the formal solution of the ODEs (13)–(15) is obtained as follows [33]. The volume $V_{\mathrm{tot}}$ of the ROI is partitioned as
(19)$$V_{\mathrm{tot}}=V_{\mathrm{int}}+V_{\mathrm{cyt}}+V_{\mathrm{er}}+V_{\mathrm{blood}}\;,$$
where $V_{\mathrm{blood}}$ and $V_{\mathrm{int}}$ denote the volume occupied by blood and interstitial fluid, respectively; $V_{\mathrm{cyt}}$ and $V_{\mathrm{er}}$ denote the total volumes of cytosol and ER in tissue cells. The total activity $V_{\mathrm{tot}}C_{T}$ in $V_{\mathrm{tot}}$ is related to the state variables and the IF by
(20)$$V_{\mathrm{tot}}C_{T}=V_{\mathrm{int}}C_{f}+V_{\mathrm{cyt}}C_{f}+V_{\mathrm{cyt}}C_{p}+V_{\mathrm{er}}C_{r}+V_{\mathrm{blood}}C_{b}\;.$$

The volume fractions of blood and interstitial fluid are defined as
(21)$$V_{b}=\frac{V_{\mathrm{blood}}}{V_{\mathrm{tot}}}\;,\;V_{i}=\frac{V_{\mathrm{int}}}{V_{\mathrm{tot}}}\;,$$
and the further dimensionless parameter $v_{r}$ as
(22)$$v_{r}=\frac{V_{\mathrm{er}}}{V_{\mathrm{cyt}}+V_{\mathrm{er}}}.$$

Notice that $v_{r}$ is independent of the number of cells and may be estimated as the ratio of the ER volume and the sum of the cytosolic and reticular volumes of any cell. It is found that
(23)$$\frac{V_{\mathrm{er}}}{V_{\mathrm{tot}}}=v_{r}\left(1-V_{b}-V_{i}\right)\;,\;\frac{V_{\mathrm{cyt}}}{V_{\mathrm{tot}}}=\left(1-v_{r}\right)\left(1-V_{b}-V_{i}\right)\;.$$

Accordingly, Equation (Equation 20) may be written in the equivalent form
(24)$$C_{T}=\alpha_{1}\;C_{f}+\alpha_{2}\;C_{p}+\alpha_{3}\;C_{r}+V_{b}\;C_{b}\;,$$
which is the IPE of 3-CM and where the positive dimensionless constants $\alpha_{1}$, $\alpha_{2}$, and $\alpha_{3}$ are defined as
(25)$$\begin{array}{c} \alpha_{1}=V_{i}+\left(1-v_{r}\right)\left(1-V_{b}-V_{i}\right)\;, \end{array}$$
(26)$$\begin{array}{c} \alpha_{2}=\left(1-v_{r}\right)\left(1-V_{b}-V_{i}\right)\;, \end{array}$$
(27)$$\begin{array}{c} \alpha_{3}=v_{r}\left(1-V_{b}-V_{i}\right)\;. \end{array}$$

**Remark** **1.**
*As intentionally suggested by similarities in notation, 2-CM is a simplification of 3-CM. It has already been observed that the rate constant $k_{6}$ in 3-CM corresponds to $k_{4}$ in 2-CM, in that both parameters are related to the process of dephosphorylation of the tracer molecules.*


More in general, there are various possibilities of analyzing the correspondence between models from a formal viewpoint. For example, consider the system of ODE for 3-CM and suppose that $C_{p}$ is almost constant. Indeed, repeated applications of 3-CM have shown that $C_{p}$ reaches a stationary value after a very short transition time [13,33]. In that case, Equation (14) reduces to $k_{3}\;C_{f}=k_{5}\;C_{p}$ (as a consequence, $C_{f}$ also becomes constant, consistent with simulations). After substitution of this condition, (14) takes the form
(28)$${\dot{C}}_{r}=k_{3}\;C_{f}-k_{6}\;C_{r}.$$Equations (Equation 13) and (Equation 28) coincide with (Equation 4) and (5) of 2-CM, provided $C_{r}$ and $k_{6}$ are identified with $C_{p}$ and $k_{4}$, respectively. This is also consistent with the observation that accumulation of FDG occurs in ER, rather than cytosol [13,33]. In a sense, a stable amount of phosphorylated tracer remains in cytosol, providing a similarly stable flux $k_{5}\;C_{p}=k_{3}\;C_{f}$ towards ER. A further point to consider while connecting the two models is related to the fact that phosphorylated tracer in 2-CM occupies the volume $V_{\mathrm{cyt}}$, while $C_{r}$ in 3-CM occupies the volume $V_{\mathrm{er}}$.

### 3.4. Compact Formulation and General Formal Solution of the Direct Problem

We describe an alternative formulation of the ODEs for CMs which is given in matrix form, and is used in a number of developments. This compact formulation writes
(29)$$\dot{C}=M\;C+k_{1}\;C_{b}\;e\;.$$

In this equation, C is the *n*-dimensional column vector made of the tracer concentrations in each compartment; M is a square matrix of order *n*, with constant entries given by the rate coefficients; e is a constant *n*-dimensional normalized column vector. Combining Equation (Equation 29) and the initial condition C(0)=0 leads to a Cauchy problem for the unknown state vector C. We refer to its solution as the solution of the direct problem.

For 2-CM, comparison with (Equation 4), (5) shows that
(30)$$M=\left[\begin{array}{cc} -(k_{2}+k_{3}) & k_{4}\\ k_{3} & -k_{4} \end{array}\right],\;C=\left[\begin{array}{c} C_{f}\\ C_{p} \end{array}\right],\;e=\left[\begin{array}{c} 1\\ 0 \end{array}\right]\;.$$

Notice that the matrix M is singular for irreversible models; conversely, M is non-singular with eigenvalues $\lambda_{1}$ and $\lambda_{2}$ from Equation (Equation 11) for reversible models. Similarly, the system (Equation 13)–(15) for 3-CM is written in the form (Equation 29) with
(31)$$M=\left[\begin{array}{ccc} -(k_{2}+k_{3}) & 0 & k_{6}\\ k_{3} & -k_{5} & 0\\ 0 & k_{5} & -k_{6} \end{array}\right],\;C=\left[\begin{array}{c} C_{f}\\ C_{p}\\ C_{r} \end{array}\right],\;e=\left[\begin{array}{c} 1\\ 0\\ 0 \end{array}\right].$$

The general structure and properties of the system matrix M have been extensively discussed, e.g., in References [34,35]. We only observe that, if the system is irreversible, then there is at least one compartment, say $C_{n}$, such that the tracer cannot get out. This is equivalent to the statement that the diagonal element $M_{nn}$ of the matrix M vanishes. As an immediate consequence of (Equation 30) and (Equation 31), the system matrix of irreversible 2-CMs and 3-CMs is singular. The result holds in general for irreversible CMs.

The solution of the direct problem can be written as
(32)$$C\left(t;k,C_{b}\right)=k_{1}\;\int_{0}^{t}e^{M\;(t-\tau )}\;\;\;C_{b}\left(\tau \right)\;d\tau \;e,$$
where $k={\left(k_{1},k_{2},\ldots ,k_{m}\right)}^{T}$ is the vector whose components are the microparameters, with upper *T* denoting the transpose, while M depends on the specific CM. We adopted a notation that points out the dependence of C on the rate constants and the input function.

The compact form of the IPE is given by
(33)$$C_{T}=\alpha \;C\left(t;k,C_{b}\right)+V_{b}\;C_{b},$$
where α is a constant row vector of order *n*, with components possibly depending on the physiological parameters. Equation (Equation 33) reduces to (Equation 12) for 2-CM if $\alpha =[1-V_{b},1-V_{b}]$. Similarly, we recover Equation (Equation 24) for 3-CM by letting $\alpha =[\alpha_{1},\alpha_{2},\alpha_{3}]$.

Replacing expression (Equation 32) of C into (Equation 33) provides the IPE governing the inverse problem for the unknown vector k, at given $C_{T}$ and $C_{b}$.

**Remark** **2.**
*In the present approach, we have described a rather simple and essential formulation of compartmental analysis, which however is sufficient to deal with standard applications in nuclear medicine. In more general situations, tracer could be delivered from blood to more than one compartment, with different time rate constants, which implies a change in the definition of the vector e. Furthermore, we have assumed that the observed activity results from the superposition of the activities of all tissue compartments, but it may happen that more than one, say h outputs, could be observed, so that the IPE should be replaced by a system of h equations.*


## 4. Patlak and Logan Graphical Approaches

Graphical methods are essentially based on the following observation. Two appropriate functions of the measurements identify the parametric representation of a time dependent plane curve which becomes linear for large time values. Application of a linear regression method to the plot provides an estimate of a corresponding slope, which can be given an interpretation in terms of either tracer absorption (irreversible CMs) or tracer distribution (reversible CMs). In other words, graphical methods may be regarded as procedures for the solution of a simplified inverse problem, yielding useful information on the overall tracer kinetics.

Here we describe the Patlak graphical approach (PGA) and the Logan graphical approach (LGA) which apply to irreversible and reversible CMs, respectively [36,37]. The procedure leading to the generation of the asymptotically linear curves follows from the properties of the specific CM, but the final parametric equation of the curve depends only on the available data. Therefore, we first describe the algorithm for the construction of the (asymptotically linear) curves for irreversible CMs, and we show how it works by application to a 2-CM and a 3-CM. A general approach to reversible CMs is then proposed, with specific examples.

### 4.1. PGA

PGA provides a useful consequence of the IPE holding for a family of irreversible CMs, in that it is used to estimate the net influx rate of radiotracer at large time values. Subsequent multiplication by the so-called lumped constant provides an estimate of the rate of glucose uptake [13]. The general process for deriving PGA is described in the following procedure, which is divided into three steps, for the sake of clarity. Then, applications to 2-CM and 3-CM will be described.
**First step.** The vector solution C of the irreversible system of ODE (Equation 29) is substituted into the IPE (Equation 33), which is then divided by $C_{b}$. The resulting equation takes the form
(34)$$\frac{C_{T}}{C_{b}}=\alpha_{P}\;\frac{\int_{0}^{t}C_{b}}{C_{b}}+\beta_{P}\left(t\right),$$(see the Appendix A) where $\alpha_{P}$ [min${}^{-1}$] is a constant macroparameter [25], and $\beta_{P}$ depends on $C_{b}$ and the components of C.**Second step.** In a number of relevant cases it may be shown that $\beta_{P}$ is asymptotically constant. Then, in the plane referred to Cartesian coordinates (x,y), define the functions x(t) and y(t) by
(35)$$x\left(t\right)=\frac{\int_{0}^{t}C_{b}}{C_{b}},\;y\left(t\right)=\frac{C_{T}}{C_{b}},$$t∈(0,∞). The points (x(t),y(t)) give the parametric representation of a curve which is known as the standard Patlak plot [38]. Comparison with Equation (Equation 34) and the condition on $\beta_{P}$ show that the curve is asymptotically linear. Thus the slope $\alpha_{P}$ and the adimensional constant intercept $\beta_{P}$ are estimated in terms of the data by linear regression [38]. The procedure may be applied pixel-wise.**Third step.** The interpretation of $\alpha_{P}$ is achieved by comparison with the stationary solution of the system of ODEs (Equation 29), corresponding to a constant IF. It is shown that $\alpha_{P}$ measures the rate of tracer uptake by the tissue at stationary conditions.

The PGA is illustrated with an example in Figure 3. The input function shown in Figure 3A has been taken from the public repository (https://github.com/theMIDAgroup/BCM_CompartmentalAnalysis.git, last accessed date to the website: 5th of August 2021) accompanying the publication [33], while the total concentration depicted in Figure 3B has been computed by solving the forward model of a 2-CM described by Equations (Equation 4) and (5). To this end, we set $V_{b}=0.15$, $k_{1}=0.29$ min${}^{-1}$, $k_{2}=0.66$ min${}^{-1}$, $k_{3}=0.18$ min${}^{-1}$, according to the results shown in [33], and $k_{4}=0$ min${}^{-1}$ so to have an irreversible CM as required by the PGA. As shown in Figure 3C, the Patlak plot tends asymptotically to a line.

The general proof of the procedure is given in the Appendix A. Here we show how and why it works by considering in detail irreversible 2-CMs and 3-CMs.

#### 4.1.1. PGA for 2-CM

Consider an irreversible 2-CM system: the concentrations $C_{f}$ and $C_{p}$ are given by (Equation 6) and (Equation 7), so that $C_{f}+C_{p}$ takes the form
(36)$$C_{f}+C_{p}=\frac{k_{1}\;k_{3}}{k_{2}+k_{3}}\;\int_{0}^{t}C_{b}\;d\tau +\frac{k_{2}}{k_{2}+k_{3}}\;C_{f}\;.$$

Replacing this expression into the IPE (Equation 12)and then dividing by $C_{b}$, leads to (Equation 34) with
(37)$$\alpha_{P2}=\left(1-V_{b}\right)\frac{k_{1}\;k_{3}}{k_{2}+k_{3}},$$
and
(38)$$\beta_{P2}\left(t\right)=\left(1-V_{b}\right)\;\frac{k_{2}}{k_{2}+k_{3}}\;\frac{C_{f}}{C_{b}}+V_{b},$$
where the suffix P2 refers to the PGA for the 2-CM.

Following Step 2, we observe that, if the ratio $C_{f}/C_{b}$ is asymptotically constant, then the adimensional quantity $\beta_{P2}$ is constant, as well, for large *t* values, and the Patlak plot is asymptotically linear. It may be shown that constancy of $C_{f}/C_{b}$ occurs in two remarkable cases: when the IF $C_{b}$ is (asymptotically) constant or exponentially decaying. The latter condition may be regarded as typical for tracer concentration in plasma [22].

According to Step 3, the direct interpretation of $\alpha_{P2}$ is obtained as follows. Consider the system (Equation 4) and (5) in the irreversible case ($k_{4}=0$), suppose that the IF $C_{b}$ is constant, and look for stationary solutions. Denote by an upper star the constant values. From the system of ODES it follows that $C_{f}$ and ${\dot{C}}_{p}$ assume constant values. Specifically, we find
(39)$$C_{f}^{*}=\frac{k_{1}}{k_{2}+k_{3}}\;C_{b}^{*},\;{\dot{C}}_{p}^{*}=\frac{k_{1}\;k_{3}}{k_{2}+k_{3}}\;C_{b}^{*}.$$Replacing expression (Equation 39) of ${\dot{C}}_{p}^{*}$ in the time derivative of the IPE (Equation 12), and comparing the result with the definition of $\alpha_{P2}$ shows that
(40)$${\dot{C}}_{T}^{*}=\alpha_{P2}\;C_{i}^{*}.$$

We conclude that $C_{T}$ grows at the constant rate $\alpha_{P2}\;C_{i}^{*}$. In other words, $\alpha_{P2}\;C_{i}^{*}$ represents the net tracer rate uptaken by the tissue in stationary conditions; it may be estimated directly from data, without explicit knowledge of the values of the rate constants. A pixel by pixel evaluation is also allowed. Finally, we remark that the rate of FDG uptake has been used to estimate glucose uptake through multiplication by the lumped constant [9,13,29].

**Remark** **3.**
*The coefficient $\alpha_{P}$ is obtained by re-writing the IPE (Equation 33) in the form of the Patlak plot. Since in general (Equation 33) depends on the volume fraction, the same holds for $\alpha_{P2}$, as brought into evidence by definition (Equation 37). The slope coefficient $\alpha_{P2}$, which is related to the rate of tracer uptake, comes from application of linear regression to the whole Patlak curve, with t varying from 0 to ∞. The coefficient $\alpha_{2P}$ is different from the slope of the line asymptotically approximating the Patlak curve [38]. This remark holds independently of the number of compartments.*


**Remark** **4.**
*A required condition for application of PGA is that $k_{4}=0$. Nevertheless, it is rather common to use PGA for 2-CMs in order to estimate the rate of tracer uptake. To discuss here the feasibility of this approach, we look at IPE for small values of $k_{4}$. We show that the sum $C_{f}+C_{p}$ converges to the expression of the irreversible case ($k_{4}=0$) for $k_{4}\to 0$. In other words, the reversible model converges to the irreversible one, as to the formulation of IPE. Thus, the value of the accumulation rate obtained by application of the irreversible model (which coincides with $\alpha_{P2}$) may be interpreted as an approximation of the overall tracer uptake rate for small $k_{4}$.*


Suppose now that $k_{4}\neq 0$. According to (Equation 8) and (Equation 9), we have
$$C_{f}+C_{p}=\frac{k_{1}}{\lambda_{1}-\lambda_{2}}\;\left[\left(k_{3}+k_{4}+\lambda_{1}\right)\;I_{1}-\left(k_{3}+k_{4}+\lambda_{2}\right)\;I_{2}\right]\;.$$

Further, accounting for the definitions, we find that
$$\lambda_{1}\to 0,\;\lambda_{2}\to -\left(k_{2}+k_{3}\right),\;\lambda_{1}-\lambda_{2}\to \left(k_{2}+k_{3}\right)\;,$$
$$I_{1}\to \int_{0}^{t}C_{b},\;k_{1}\;I_{2}\to k_{1}\;\int_{0}^{t}e^{-\left(k_{2}+k_{3}\right)\;\left(t-\tau \right)}C_{b}\left(\tau \right)\;d\tau =C_{f,k_{4}=0}$$
for $k_{4}\to 0$; in particular $C_{f,k_{4}=0}$ refers to the expression (Equation 6) of $C_{f}$, evaluated at $k_{4}=0$. It follows that
$$C_{f}+C_{p}\to \frac{k_{1}\;k_{3}}{k_{2}+k_{3}}\;\int_{0}^{t}C_{b}\;d\tau +\frac{k_{2}}{k_{2}+k_{3}}\;C_{f,k_{4}=0}\;\left(k_{4}\to 0\right),$$
which corresponds to Equation (Equation 36) and which leads to the Patlak plot. Thus the estimate of the rate of tracer uptake $\alpha_{P2}$ by application of the PGA is taken as an approximate estimate of the rate in the presence of a small dephosphorylation effect.

#### 4.1.2. PGA for 3-CM

Consider now the irreversible 3-CM. To accomplish Step 1, Equation (Equation 24) is re-written in the equivalent form
(41)$$C_{T}=\left(\alpha_{1}-\alpha_{3}\right)\;C_{f}+\left(\alpha_{2}-\alpha_{3}\right)\;C_{p}+\alpha_{3}\left(C_{f}+C_{p}+C_{r}\right)+V_{b}C_{b}.$$

Next, the sum of the differential Equations (Equation 13)–(15) is re-written as
$$\frac{d}{dt}\left(C_{f}+C_{p}+C_{r}\right)=-k_{2}\;C_{f}+k_{1}\;C_{b}\;.$$

Evaluation of the integral from 0 to *t*, and substitution of the explicit expression of $C_{f}$ from (Equation 16) leads to
(42)$$C_{f}+C_{p}+C_{r}=\frac{k_{1}\;k_{3}}{k_{2}+k_{3}}\;\int_{0}^{t}C_{b}+\frac{k_{2}}{k_{2}+k_{3}}\;C_{f}\;.$$

Replacing the expression (Equation 42) into (Equation 41), and dividing by $C_{b}$ leads to the standard equation for the Patlak plot (Equation 34), where
(43)$$\alpha_{P3}=\alpha_{3}\;\frac{k_{1}\;k_{3}}{k_{2}+k_{3}}\;,$$
(44)$$\beta_{P3}\left(t\right)=\left(\alpha_{1}-\alpha_{3}+\frac{k_{2}}{k_{2}+k_{3}}\right)\;\frac{C_{f}}{C_{b}}+\left(\alpha_{2}-\alpha_{3}\right)\;\frac{C_{p}}{C_{b}}+V_{b}\;.$$

According to Step 2, if $\beta_{P3}$ is asymptotically constant then an asymptotically linear Patlak plot is obtained.

Step 3 provides the interpretation of $\alpha_{P3}$. In fact, consider the system of ODEs (Equation 13)–(15), under the assumption of irreversibility ($k_{6}=0$) and look for stationary solutions at constant $C_{b}=C_{b}^{*}$. It is found, in particular, that
$${\dot{C}}_{r}^{*}=\frac{k_{1}\;k_{3}}{k_{2}+k_{3}}\;C_{b}^{*}.$$

Evaluation of the asymptotic value of the time derivative of the IPE (Equation 24) shows that ${\dot{C}}_{T}^{*}=\alpha_{3}\;{\dot{C}}_{r}^{*}$. Substitution of the previous expression of ${\dot{C}}_{r}^{*}$ and comparison with the definition of $\alpha_{P3}$ leads to
(45)$${\dot{C}}_{T}^{*}=\alpha_{P3}\;C_{b}^{*}.$$In other words, $C_{T}$ grows at the constant rate $\alpha_{P3}\;C_{b}^{*}$.

**Remark** **5.**
*Equation (Equation 45) for 3-CM is the analog of (Equation 40) for 2-CM. This is consistent with the general discussion of PGA for irreversible CMs of generic order n, which is given in the Appendix A.*
As exemplified by Equations (Equation 37) and (Equation 43), the explicit form of the equation connecting the general slope $\alpha_{P}$ to the rate constants k depends on the structure of the CM. Indeed, PGA is model-independent, but requires the use of CMs for its motivation, development, and proof.

### 4.2. LGA

Under suitable assumptions, LGA provides a useful consequence of IPE, which holds for reversible CMs; specifically, we obtain the estimate of a macroparameter representing a ratio between equilibrium concentrations. The procedure is somehow similar to that of Section 4.1 for PGA; we reproduce here the main steps in the case of a general reversible CM [7,20,37].
First step. Consider the integral in time of the IPE equation in the compact form (Equation 33):
(46)$$\int_{0}^{t}C_{T}=\int_{0}^{t}\alpha \;C+V_{b}\;\int_{0}^{t}C_{b}.$$
where αC is related to data through the ODE (Equation 29). Indeed, multiplication of (Equation 29) by $M^{-1}$ yields
$$C=M^{-1}\;\dot{C}-k_{1}\;M^{-1}\;e\;C_{b}.$$Further multiplication by α, and integration with respect to time from 0 to *t* gives
$$\int_{0}^{t}\alpha \;C\left(\tau ;k\right)\;d\tau =\alpha \;M^{-1}C-k_{1}\;\alpha \;M^{-1}\;e\;\int_{0}^{t}C_{b}\;d\tau$$Substitution into (Equation 46), followed by division by $C_{T}$ gives the necessary condition
(47)$$\frac{\int_{0}^{t}C_{T}}{C_{T}}=\alpha_{L}\;\frac{\int_{0}^{t}C_{b}}{C_{T}}+\beta_{L}\left(t\right)$$
where the dimensionless constant $\alpha_{L}$ and the function $\beta_{L}\left(t\right)$ [min${}^{-1}]$ are defined as
(48)$$\alpha_{L}=-k_{1}\;\alpha \;M^{-1}\;e+V_{b},$$
(49)$$\beta_{L}\left(t\right)=\frac{\alpha \;M^{-1}C}{C_{T}}.$$Second step. Following the analogy with the GPA approach of Section 4.1, in a number of relevant cases it may be shown that $\beta_{L}$ is asymptotically constant. In such a case, consider the functions x(t) and y(t) defined by
(50)$$x\left(t\right)=\frac{\int_{0}^{t}C_{b}}{C_{T}},\;y\left(t\right)=\frac{\int_{0}^{t}C_{T}}{C_{T}},$$t∈(0,∞). In analogy with (Equation 34) the points (x(t),y(t)) of the Cartesian plane define a parametric representation of the standard Logan plot, which is an asymptotically linear curve. The adimensional slope $\alpha_{L}$ and the intercept $\beta_{L}$ are macroparameters determined by the data, which are estimated by linear regression.Third step. The interpretation of $\alpha_{L}$ follows from the equilibrium solution of the system (Equation 29) at constant IF $C_{b}^{*}$. The equilibrium state $C^{*}$ is given by
$$C^{*}=-k_{1}\;M^{-1}\;e\;C_{b}^{*}.$$According to the IPE (Equation 33), the previous equation, and the definition of $\alpha_{L}$, it follows that
(51)$$C_{T}^{*}=\alpha \;C^{*}+V_{b}\;C_{b}^{*}=-k_{1}\;\alpha \;M^{-1}\;e\;C_{b}^{*}+V_{b}\;C_{b}^{*}=\alpha_{L}\;C_{b}^{*}$$Thus, the slope $\alpha_{L}=C_{T}^{*}/C_{b}^{*}$ is the ratio between the constant equilibrium value of the total tissue concentration and the blood concentration.

Figure 4 shows an example of the application of the LGA. The input function plotted in panel (A) is the same as in Figure 3A, while the total concentration plotted in panel (B) has been computed by solving Equations (Equation 4) and (5) or the forward model of a 2-CM where we set $V_{b}=0.15$, $k_{1}=0.29$ min${}^{-1}$, $k_{2}=0.66$ min${}^{-1}$, $k_{3}=0.18$ min${}^{-1}$ according to the results shown in [33]. To satisfy the reversibility condition, we set $k_{4}$ to a value close to that of $k_{3}$, namely we defined $k_{4}=0.3$ min${}^{-1}$. As can be seen in Figure 4C, the Logan plot tends asymptotically to a line.

**Remark** **6.**
*It should be noticed that $C_{T}^{*}$ takes into account the total tracer content of a tissue volume, which means that the contribution due to the presence of blood is also considered.*
An alternative interpretation of $\alpha_{L}$ in terms of volumes is obtained as follows. Suppose $V_{T}$ and $V_{b}$ are tissue and blood volumes containing the same amount of radioactivity at the equilibrium concentrations. This means that
(52)$$C_{T}^{*}\;V_{T}=C_{b}^{*}\;V_{b}\;,$$
whence, it follows that
(53)$$\frac{V_{b}}{V_{T}}=\frac{C_{T}^{*}}{C_{b}^{*}}=\alpha_{L}\;.$$Accordingly, $\alpha_{L}$ may also be interpreted as the ratio between blood and tissue volumes containing the same amount of tracer at equilibrium concentrations. We conclude that $\alpha_{L}$ assesses the overall capability of the tissue to concentrate or dilute tracer in equilibrium conditions.

#### 4.2.1. LGA for 2-CM

We consider a reversible 2-CM system ($k_{4}\neq 0$). Explicit computations show that
$$k_{1}\;\alpha \;M^{-1}\;e=-\left(1-V_{b}\right)\;\frac{k_{1}}{k_{2}}\;\left(1+\frac{k_{3}}{k_{4}}\right)$$
and
$$\alpha \;M^{-1}\;C=-\left(1-V_{b}\right)\;\left(\frac{k_{3}+k_{4}}{k_{2}\;k_{4}}\;C_{f}+\frac{k_{2}+k_{3}+k_{4}}{k_{2}\;k_{4}}\;C_{m}\right).$$

Substitution into the definitions (Equation 54) and (Equation 55) leads to
(54)$$\alpha_{L2}=\left(1-V_{b}\right)\;\frac{k_{1}}{k_{2}}\;\left(1+\frac{k_{3}}{k_{4}}\right)+V_{b},$$
and
(55)$$\beta_{L2}\left(t\right)=-\left(1-V_{b}\right)\;\left[\frac{k_{3}+k_{4}}{k_{2}\;k_{4}}\;\frac{C_{f}}{C_{T}}+\frac{k_{2}+k_{3}+k_{4}}{k_{2}\;k_{4}}\;\frac{C_{p}}{C_{T}}\right],$$
where the low index L2 refers to the Logan plot for 2-CMs.

Suppose that $V_{b}$ is small enough to be negleted. Then, $\alpha_{L2}$ reduces to $\left(k_{1}/k_{2}\right)\;\left(1+k_{3}/k_{4}\right)=DV_{T}$, which is called the total distribution volume [20,27,39].

#### 4.2.2. LGA for 3-CM

Consider a reversible 3-CM ($k_{6}\neq 0$). Following the same procedure as Section 4.2.1, with M, α, and e properly updated, it is found that
(56)$$\frac{\int_{0}^{t}C_{T}}{C_{T}}=\alpha_{L3}\;\frac{\int_{0}^{t}C_{b}}{C_{T}}+\beta_{L3}\left(t\right)$$
with
(57)$$\alpha_{L3}=\alpha \;M^{-1}\;e+V_{b}=\frac{k_{1}}{k_{2}}\;\left(\alpha_{1}+\alpha_{2}\;\frac{k_{3}}{k_{5}}+\alpha_{3}\;\frac{k_{3}}{k_{6}}\right)+V_{b}$$
(58)$$\beta_{L3}\left(t\right)=\frac{\alpha \;M^{-1}\;C}{C_{T}}=-\left(\frac{\alpha_{1}}{k_{2}}+\frac{\alpha_{2}\;k_{3}}{k_{2}\;k_{5}}+\frac{\alpha_{3}\;k_{3}}{k_{2}\;k_{6}}\right)\;\frac{C_{f}+C_{p}+C_{r}}{C_{T}}-\frac{\alpha_{2}}{k_{5}}\;\frac{C_{p}}{C_{T}}-\frac{\alpha_{3}}{k_{6}}\;\frac{C_{p}+C_{r}}{C_{T}}$$

If $\beta_{L3}\left(t\right)$ is asymptotically constant, then Equation (Equation 56) provides a standard Logan plot with slope $\alpha_{L3}$.

An analysis of the system (Equation 13)–(15) and Equation (Equation 24) at constant values of $C_{f}$, $C_{p}$, $C_{r}$, and $C_{b}$ shows that $\alpha_{L,3C}$ coincides with $C_{T}^{*}/C_{b}^{*}=V_{b}/V_{T}$ such that $C_{T}^{*}\;V_{T}=C_{b}^{*}\;V_{b}.$

## 5. Issues on the Solvability of the Inverse Problem

The fundamental application of compartmental analysis deals with the reconstruction of tracer kinetics via estimate of the rate constants, which cannot be measured directly. A natural requirement is that the parameters that identify the mathematical model are uniquely defined, up to noise in the data. This is known as the identifiability problem. There are, at least, two reasons to asses identifiability [40]. Since the rate constants have a kinetic meaning, we are interested in knowing whether their values can be determined uniquely from the available experimental data. Moreover, we expect difficulties in the estimate of parameters of a non-identifiable model. The discussion of a few aspects of these problems is the main aim of the present section with a particular focus on linear compartment models. The analysis of identifiability for nonlinear compartmental systems, such as those arising when fluxes between compartments are modeled by the Michaelis-Menten law is still an open problem. A comparison of currently available techniques for nonlinear model is beyond the scope of this review and can be found in Reference [40].

### 5.1. Identifiability of Linear CMs

In some scenarios, many distinct models may exist equally fitting the recorded data [33]. Here, we assume that a linear compartmental model has been fixed by exploiting, e.g., some a priori information on the biochemical process under consideration. Additionally, we assume all the physiological parameters, such as the blood volume fraction $V_{b}$, to be known, and we investigate the identifiability of the kinetic parameters k.

A standard approach to discuss the identifiability of linear CMs consists in computing the Laplace transform of both sides of the IPE (Equation 33) and of the system of ODEs (Equation 29), leading to, respectively,
(59)$${\tilde{C}}_{T}\left(s\right)=\alpha \;\tilde{C}\left(s\right)+V_{b}\;{\tilde{C}}_{b}\left(s\right)\;,$$
and
(60)$$\left(s\;I-M\right)\;\tilde{C}\left(s\right)=k_{1}\;{\tilde{C}}_{b}\left(s\right)\;e\;,$$
where we have indicated by $\tilde{f}\left(s\right)$ the Laplace transform of a function f(t), and we have assumed suitable regularity conditions.

Provided the matrix ${\left(s\;I-M\right)}^{-1}$ is invertible, by computing the solution $\tilde{C}\left(s\right)$ of the linear system (Equation 60) and substituting it into Equation, (Equation 59) we obtain
(61)$$\frac{{\tilde{C}}_{T}\left(s\right)-V_{b}\;{\tilde{C}}_{b}\left(s\right)}{{\tilde{C}}_{b}\left(s\right)}=k_{1}\alpha {\left(s\;I-M\right)}^{-1}e=k_{1}\frac{Q(s,\hat{k})}{D(s;\hat{k})}\;,$$
where $\hat{k}={\left[k_{2},\ldots ,k_{m}\right]}^{T}$, $D\left(s;\hat{k}\right):=\det\left(s\;I-M\right)$ is a polynomial of degree *n* in the variable *s* with coefficients depending on $\hat{k}$, and similarly $Q\left(s;\hat{k}\right):=\alpha \;\mathrm{adj}\left(s\;I-M\right)\;e$ is a polynomial of degree up to n−1, adj(sI−M) being the adjugate of the n×n matrix (sI−M). In the following, we assume that $Q(s;\hat{k})$ and $D(s;\hat{k})$ are coprime and that they have a leading coefficient equal to 1.

Any alternative set $h={\left[h_{1},\ldots ,h_{m}\right]}^{T}$ of kinetic parameters that satisfies the IPE (Equation 33) and the system of ODEs (Equation 29) for the recorded input function $C_{b}\left(t\right)$ and total concentration $C_{T}\left(t\right)$ must satisfy Equation (Equation 61), with k replaced by h. Since the left-side of Equation (Equation 61) does not depends on the value of the kinetic parameters, the following necessary condition holds:(62)$$k_{1}\;\frac{Q(s,\hat{k})}{D(s;\hat{k})}=h_{1}\;\frac{Q(s,\hat{h})}{D(s;\hat{h})}.$$

By exploiting the fact that $Q(s,\hat{k})$ and $D(s;\hat{k})$ have leading coefficients equal to 1, it can be easily shown that Equation (Equation 62) implies $k_{1}=h_{1}$. Then, since $Q(s,\hat{k})$ and $D(s;\hat{k})$ are coprime, from Equation (Equation 62) it follows that
(63)$$Q\left(s,\hat{h}\right)=Q\left(s;\hat{k}\right),\;D\left(s,\hat{h}\right)=D\left(s;\hat{k}\right).$$

If these two equations imply $\hat{h}=\hat{k}$, then identifiability is proved. The actual implementation of this last step depends on the particular CM under consideration and may be rather involved when many compartments are included in the model.

As an illustrative example, we consider a reversible 2-CM. In this case,
(64)$$Q\left(s;\hat{k}\right)=s+k_{3}+k_{4}$$
and
(65)$$D\left(s;\hat{k}\right)=s^{2}+\left(k_{2}+k_{3}+k_{4}\right)s+k_{2}k_{4}\;,$$
which do not have any common roots and have a leading coefficient equal to 1. In this case, the necessary conditions in Equation (Equation 63) lead to the system
(66)$$\left\{\begin{array}{c} k_{3}+k_{4}=h_{3}+h_{4}\\ k_{2}+k_{3}+k_{4}=h_{2}+h_{3}+h_{4}\\ k_{2}k_{4}=h_{2}h_{4} \end{array}\right.$$
that straightforwardly implies $\hat{k}=\hat{h}$.

Some additional results on the identifiability of 2-CMs and 3-CMs can be found in Reference [41] where more general models are also consider, including, e.g., multiple compartments exchanging a tracer with blood.

### 5.2. Sensitivity Analysis

In the previous section, we showed how to analytically prove the identifiability of the rate constants of a linear CM. However, in practical scenarios, this analytic result does not guarantee that the rate constants may be effectively and reliably estimated. This may occur when changes in one or more parameters only slightly affect the data. For this reason, a local (or global) sensitivity analysis needs to be performed to investigate to what extent the state of the system changes when parameter values are perturbed from a reference value [42,43,44]. For example, the Matlab code for performing a local sensitivity analysis of a 3-CM is provided in the github repository accompanying [33]. More comprehensive toolboxes implementing various global sensitivity analysis approaches are available either in Matlab [45] or Python [46].

While for standard 2-CMs local sensitivity analysis shows that perturbations of the rate constants significantly affect the system variables, the 3-CM show a poor sensitivity with respect to $k_{5}$ and $k_{6}$ [33]. As a consequence, even though the parameters are identifiable, multiple configurations exist that equally fit the noisy recorded data. The uniqueness of the solution of the inverse problem can be restored by incorporating in the optimization procedure information on biologically feasible interval for the values of such parameters available a priori [33] or derived by a previous sensitivity analysis [47].

## 6. Physiology-Driven Compartmental Models

CMs provide a highly flexible instrument which may be adapted to the analysis of tracer kinetics in non standard conditions. We examine here a few examples, focusing on the essential features of each approach. Each example accounts for a set of specific conditions and available PET data, which in turn suggest the most convenient approach to the modelling of tracer kinetics. The main characteristics of all the considered compartment models shall be summarized in Table 1.

### 6.1. Reference Tissue Models

In the present formulation, reference tissue models (RTMs) result from the combination of 1-CM, 2-CM, and a graphical approach. RTMs have been introduced to overcome difficulties in the reconstruction of the TAC of the IF. Often, the IF is determined by measurements of activity on an ROI which is drawn over a sufficiently large blood pool, such as the left ventricle, but the procedure is subject to systematic errors. Possible sources of error are given by spillover, cardiac motion, and the low resolution of PET cameras (see Reference [22] and the related references). Moreover, at the very beginning of the diffusion process the arterial TAC shows a very high peak, which is difficult to reliably estimate [48]. The essential idea of RTMs, is that the TAC of a suitably chosen reference tissue (RT) replaces the TAC of the IF of the target tissue.

We consider an RTM which is formed by a reversible 2-CM for the TT, and a 1-CM for the RT, as shown in Figure 5. In particular, $C_{R}$ denotes tracer concentration in the RT, while $k_{1R}$ and $k_{2R}$ [min${}^{-1}$] are the rate constants for incoming tracer flow from blood, and outgoing flow to blood, respectively. Thus, the RTM depends on six unknown kinetic parameters. The natural data are the time dependent radioactivity concentration $C_{R}$ of the RT, and total concentration $C_{T}$ of the target tissue (TT). In the present formulation we also assume that the concentration $C_{b}$ of blood is known from $t_{0}$ on, with $t_{0}$ sufficiently large. We show that the IPE for the unknown rate constants may be formulated in terms of these data.

Following the approach of Reference [48] we firstly describe the RT. The concentration $C_{R}$ solves the Cauchy problem for the 1-CM:(67)$${\dot{C}}_{R}=-k_{2R}\;C_{R}+k_{1R}\;C_{b},\;C_{R}\left(0\right)=0\;.$$

We denote by $V_{bR}$ the given volume fraction of RT. Then the measured total radioactivity concentration $C_{R}$ is expressed as
(68)$$C_{R}=\left(1-V_{bR}\right)\;C_{R}+V_{bR}\;C_{b}.$$

As to the TT, this is represented as a 2-CM. Hence Equation (Equation 29) applies, with M, C, and e given by (Equation 30). It follows that
(69)$$C=k_{1}\;\int_{0}^{t}e^{M\;(t-\tau )}\;\;\;C_{b}\left(\tau \right)\;d\tau \;e\;,$$
where the unknown parameters are $\left(k_{1},k_{2},k_{3},k_{4}\right)$. Notice that Equation (Equation 69) is a particular case of the general representation (Equation 32), which has been restated here for convenience. The connection between the measured total radioactivity concentration $C_{T}$ of the TT and the related state vector C of Equation (Equation 69) follows from Equation (Equation 33), which here takes the form
(70)$$C_{T}=\alpha_{T}\;C+V_{bT}\;C_{b},\;\alpha_{T}=\left(1-V_{bT}\right)\;\left[\begin{array}{cc} 1 & 1 \end{array}\right],$$
where $V_{bT}$ is the given volume fraction of the TT.

In order to find the IPE for the RTM we proceed according to three steps.

In the first step the constant $k_{2R}$ is expressed in terms of $k_{1R}$ by the use of RT data. This reduces the number of the unknown parameters from 6 to 5, as well as and guarantees the identifiability. Specifically, we set
(71)$$k_{2R}=\lambda \;k_{1R},$$
where λ is a constant unknown parameter that may be estimated by a graphical approach. Indeed, it follows from (Equation 67) and (Equation 68) that the following identity holds:(72)$$\frac{\int_{t_{0}}^{t}C_{R}}{C_{R}}=\gamma_{1}\frac{\int_{t_{0}}^{t}C_{b}}{C_{R}},+\gamma_{2}\left(t\right)\;,$$
where
$$\gamma_{1}=\frac{1-V_{bR}}{\lambda }+V_{bR}.$$

Under the assumption that $\gamma_{2}$ is asymptotically constant, Equation (Equation 72) provides $\gamma_{1}$ by linear regression, hence λ.

In the second step $C_{b}$ is expressed in terms of $C_{R}$ as a consequence of Equations (Equation 67) and (Equation 68) for the RT. It is found that
(73)$$C_{b}=\frac{C_{R}}{V_{bR}}-\left(V_{bR}^{-1}-1\right)\;k_{1R}\int_{0}^{t}e^{-\gamma \;k_{1R}\;\left(t-\tau \right)}\;C_{R}\;d\tau \;.$$

In the third step, replacement of $C_{b}$ in (Equation 69) with its expression (Equation 73) yields the state vector C in terms of $C_{R}$. Subsequent substitution of C and $C_{b}$ in (Equation 70) provides the IPE for the five unknowns $k_{1R}$, $k_{1}$, $k_{2}$, $k_{3}$, $k_{4}$.

Identifiability is proved by considering the Laplace transforms of Equations (Equation 29) and (Equation 70). The details can be found in Reference [48].

We conclude this section with a few remarks. The volume fraction $V_{bR}$ and $V_{bT}$ are often set equal to zero. The number of the unknown parameters is reduced by the assumption that the distribution volumes of tracer of the two tissues are equal; this corresponds to imposing
$$\frac{k_{1}}{k_{2}}=\frac{k_{1R}}{k_{2R}}$$
but application of this the assumption to tumor tissues has been subject to criticism. We refer again to Reference [48] for more details.

### 6.2. CMs for Liver

In applications of compartmental analysis to the liver system there are two input functions to consider in that blood, hence the tracer, is supplied to liver by both the hepatic artery (HA) and the portal vein (PV), which carries to the liver the blood outgoing from the gut. While tracer concentration in HA may be estimated by the methods developed for standard IFs, the PV is not accessible to PET images. As observed in Reference [16], there have been several attempts to estimate the dual-input IFs from dynamic PET data. According to the approach of Reference [16], a reliable solution is provided by combination of three compartmental models and suitable physiologic remarks.

As described in Figure 6, we developed a compartmental approach resulting from the combination of two 2-CM subsystems for tracer kinetics in the gut and the liver, respectively, and a 1-CM subsystem for the portal vein, regarded as a pool connecting gut and liver.

The gut subsystem is regarded as a reversible 2-CM with arterial blood concentration for IF, and portal vein concentration for the output function. The following system of ODEs holds, which is simply a restatement of the ODEs for a 2-CM: (74)$$\begin{array}{cc} {\dot{C}}_{f}^{\prime } & =-\left(k_{vf}^{\prime }+k_{pf}^{\prime }\right)\;C_{f}^{\prime }+k_{fp}^{\prime }\;C_{p}^{\prime }+k_{fb}^{\prime }\;C_{b} \end{array}$$(75)$$\begin{array}{cc} {\dot{C}}_{p}^{\prime } & =k_{pf}^{\prime }\;C_{f}^{\prime }-k_{fp}^{\prime }\;C_{p}^{\prime }\;. \end{array}$$

Here, $C_{f}^{\prime }$ and $C_{p}^{\prime }$ denote the tracer concentrations of the free compartment $C_{f}^{\prime }$, and the phosphorylated compartment, $C_{p}^{\prime }$, respectively; $C_{b}$ is the arterial blood concentration. We remark that the notation for the rate coefficients has been changed in order to take into account the complicated structure of the model: specifically, $k_{ij}$ denotes the rate coefficient for tracer transfer to the target compartment $C_{i}$ from the source compartment $C_{j}$. In particular, $k_{vf}^{\prime }$ is related to the rate of transfer to the PV from the gut.

The total concentration $C_{T,\mathrm{gut}}$ of the gut system and the IF $C_{b}$ are accessible to measurement and hence are regarded as data for the standard compartmental problem. Thus the rate coefficients are determined by solving the IPE
$$C_{T,\mathrm{gut}}=C_{f}^{\prime }+C_{p}^{\prime },$$
where a vanishing blood volume fraction is considered. The coefficients are replaced into the system (Equation 74) and (75), so that the concentrations $C_{f}^{\prime }$ and $C_{p}^{\prime }$ are evaluated by solving the related Cauchy problem with vanishing initial data. In particular, the time course of $C_{f}^{\prime }$ is required for further developments.

The PV subsystem, denoted as $C_{v}$, provides the input concentration to liver from gut. Tracer carried by blood leaves the free gut compartment $C_{f}^{\prime }$, goes through $C_{v}$, and enters the free liver compartment $C_{f}$. While observing that the incoming and the outgoing blood flows of $C_{v}$ are constant, it is found that $k_{vf}^{\prime }=k_{fv}$. Thus, the ODE for the concentration $C_{v}$ takes the form
$${\dot{C}}_{v}=k_{vf}^{\prime }C_{f}-k_{vf}^{\prime }C_{v};$$
its formal solution yields the expression of $C_{v}$ in terms of $C_{f}^{\prime }\left(t\right)$ and the rate constant $k_{vf}^{\prime }$.

The liver subsystem is modeled as a reversible 2-CM. The ODEs for the concentrations $C_{f}$ and $C_{p}$, of free and metabolized compartments are
(76)$$\begin{array}{cc} {\dot{C}}_{f} & =-\left(k_{pf}+k_{sf}\right)\;C_{f}+k_{fp}\;C_{p}+k_{fb}\;C_{b}+k_{vf}^{\prime }C_{v}\;, \end{array}$$
(77)$$\begin{array}{cc} {\dot{C}}_{p} & =k_{pf}\;C_{f}-k_{fp}\;C_{p}\;. \end{array}$$

The interpretation of the rate coefficients follows according to the general rules, with the further remark that $k_{sf}$ is the rate towards the suprahepatic vein from the liver free pool $C_{f}$. The IFs $C_{b}$ and $C_{v}$ can now be regarded as given. The IPE takes the form
$$C_{T,\mathrm{liver}}=\left(1-V_{b}\right)\;\left(C_{f}+C_{p}\right)+V_{b}\;\left(0.11C_{b}+0.89C_{v}\right),$$
where $C_{T,\mathrm{liver}}$ is the measured concentration in liver, while the numerical coefficients 0.11 and 0.89 refer to the rate of arterial and venous contributions to the hepatic blood content $V_{b}$ per unit volume. The unknowns are the 5 rate constants $k_{pf}$, $k_{sf}$, $k_{fp}$, $k_{fb}$, $k_{vf}^{\prime }$.

### 6.3. CMs for the Renal System

Tracer subtraction from blood by the renal system, besides being of interest in itself, may be influenced by the presence of drugs, thus leading to possible therapeutic applications [14]. A quantitative analysis of the process of renal excretion involves a compartment anatomically represented by the bladder, which is thus, to be considered in the formulation of the renal CM. Moreover, the change of tracer concentration in tubules associated with re-absorption of liquid must also be considered. Insertion of these conditions makes the CM of the renal system rather complex.

A concise description of models available in the literature can be found in References [14,49]. Here we describe the CM of Reference [14], which is capable of accounting for a number of physiological conditions.

We refer to Figure 7, showing that the renal CM involves four compartments, which can be given the following interpretations:An extravascular compartment $C_{f}$ accounting for tracer outside cells, whose exchange with blood is free.A compartment $C_{p}$ containing the phosphorylated FDG, the FDG in the cells, and the preurine pool. In particular, following the flow of liquid, tracer is filtered in the preurines and carried towards the proximal tubule. This compartment has been denoted as $C_{p}$ because tracer can also be in phosphorylated form.A tubular compartment $C_{t}$, where tracer flows towards bladder. Here, the concentration varies (increases) because of the re-absorption of liquids through the tubular walls.The urinary pool $C_{u}$, anatomically identified with the bladder, where the tracer carried by the urine is accumulated. Notice the bladder volume varies with time.

Following the standard conventions, the system of ODEs may be written as follows:(78)$${\dot{C}}_{f}=-\left(k_{bf}+k_{pf}\right)\;C_{f}+k_{fp}\;C_{p}+k_{fb}\;C_{b}\;,$$
(79)$${\dot{C}}_{p}=k_{pf}\;C_{f}-\left(k_{fp}+k_{tp}\right)\;C_{p}+k_{pb}C_{b},$$
(80)$${\dot{C}}_{t}=-k_{ut}\;C_{t}+k_{tp}C_{p},$$
and
(81)$$\frac{d}{dt}\left(V_{u}C_{u}\right)=F_{ut}C_{t}.$$

Notice that $V_{u}$ denotes the time dependent volume of the bladder, so that $V_{u}C_{u}$ is the corresponding total activity content. $F_{ut}$ (mL min${}^{-1}$) denotes the bulk flow of urine from tubules to bladder; according to the assumption of stationarity, $F_{ut}$ is considered constant.

The total radioactivity $C_{K}$ of the kidneys may be written as
(82)$$C_{K}=\left(1-\eta_{b}-\eta_{t}\right)\left(C_{f}+C_{p}\right)+\eta_{t}C_{t}+\eta_{b}C_{b},$$
where $\eta_{b}$ and $\eta_{t}$ denote the fractions of kidney volume $V_{K}$ occupied by the tubular compartment and the blood compartment, respectively; they are regarded as given. The measured data are $C_{K}$, $C_{u}$, and $V_{u}$.

A rather complicated formal development (see Reference [14]) shows that, in view of (Equation 78)–(Equation 80), Equations (Equation 82) and (Equation 81) are approximated by
(83)$$A_{K}=x_{1}V_{K}C_{b}+x_{2}\int_{0}^{t}A_{K}+x_{3}V_{K}\int_{0}^{t}C_{b}+x_{4}V_{K}A_{u},$$
(84)$$A_{u}=z_{1}\int_{0}^{t}\int_{0}^{\tau }C_{k}+z_{2}\int_{0}^{t}\int_{0}^{\tau }C_{b}+z_{3}\int_{0}^{t}A_{u}.$$

Here, $A_{K}=V_{K}C_{K}$ and $A_{u}=V_{u}C_{u}$ represent the total activities of kidneys and bladder, respectively. They are considered as given, since they are expressed in terms of given data. The parameters $\left(x_{1},x_{2},x_{3},x_{4}\right)$ and $\left(z_{1},z_{2},z_{3}\right)$ depend on the rate constants $k_{ij}$, the volume fractions $\eta_{b}$ and $\eta_{t}$, and the flow parameter $F_{ut}$. System (Equation 83) and (Equation 84) provides the starting point for the solution of the inverse problem for the unknown rate constants.

The number of unknowns is reduced by the introduction of two physiological constraints.

First, the constant flux rate $F_{ut}$ into bladder can be estimated from the measured bladder volumes as
$$F_{ut}=\frac{V_{u}\left(t_{f}\right)-V_{u}\left(t^{*}\right)}{t_{t}-t^{*}}$$
where $t_{f}$ is the final time, and $t^{*}$ is any intermediate time.

Second, we recall that the bulk flow of carrier fluid from $C_{t}$ toward the bladder is around two orders of magnitude smaller than the reabsorbed flow through the boundary of $C_{t}$. Therefore, the tracer balance equation in tubule implies $k_{tp}=10^{2}k_{ut}$. Finally, it follows from the definitions that $k_{ut}\;\eta_{t}V_{K}=F_{ut}$, whence $k_{ut}$ and $k_{tp}$ follow.

A simplified version of this renal system proposed in Reference [49] provides an example of application of an inversion procedure inspired by biology. Here, the bladder volume has been regarded as constant, while the compartment $C_{t}$, which accounts for the presence of water re-absorption in tubule, has not been considered.

The ODEs of the simplified model take the form
(85)$$\begin{array}{cc} {\dot{C}}_{f} & =-\left(k_{bf}+k_{pf}\right)\;C_{f}+k_{fp}\;C_{p}+k_{fb}\;C_{b}, \end{array}$$
(86)$$\begin{array}{cc} {\dot{C}}_{p} & =k_{pf}\;C_{f}-\left(k_{fp}+k_{up}\right)\;C_{p}+k_{pb}C_{b}, \end{array}$$
(87)$$\begin{array}{cc} {\dot{C}}_{u} & =k_{up}C_{p}. \end{array}$$

The formal expressions of $C_{f}\left(t\right)$ and $C_{p}\left(t\right)$ are obtained by solving the Cauchy Problem (Equation 85) and (86) with vanishing initial conditions. Then, $C_{u}\left(t\right)$ is determined by integration in time of $k_{up}C_{p}$. The data are the total renal concentration $C_{K}$, the bladder concentration ${\bar{C}}_{u}$, and the IF $C_{p}$. The total renal concentration $C_{K}$ is related to the unknown rates by the equation
(88)$$C_{K}=\left(1-V_{b}\right)\left(C_{f}+C_{p}\right)+V_{b}C_{b}$$
while ${\bar{C}}_{u}$ and $C_{u}$ are simply related by ${\bar{C}}_{u}={\bar{C}}_{u}$.

### 6.4. The Role of the Endoplasmic Reticulum

In Section 3.3.3 we have discussed a three-compartment model that aims to account for the crucial role played by the endoplasmic reticulum in cancer FDG metabolism. This model relies on results obtained by means of an in vitro argument [13] and has been recently confirmed in vivo using data recorded by means of a PET device for animal models [33]. A scheme of the model is illustrated in Figure 2 and the corresponding equations have been discussed in Section 3.3.3.

### 6.5. Comparison among Different Models

For the convenience of the reader, the most important models examined in the previous sections are summarized in Table 1. For each of the models, all the relevant features are outlined. Applicability of the PGA has also been considered, in view of its rather general use.

## 7. Some Numerics: Optimization Schemes

The way compartmental analysis can be actually exploited for modeling the tracer kinetics in nuclear medicine relies on the numerical solution of the IPE Equation (Equation 33). In experimental applications, the input data is given by the time series $C_{T}=C_{T}\left(t\right)$, which is determined by computing the pixel content in Regions of Interest (ROIs) of the reconstructed PET data at different time points; $C_{b}=C_{b}\left(t\right)$ is the input function, which is also determined from PET data; and the unknown is represented by the vector k, whose components are the tracer coefficients, and by $V_{b}$. By denoting the right hand side of Equation (Equation 33) as
(89)$$F\left(k,V_{b}\right):=\alpha \;C\left(t;k,C_{b}\right)+V_{b}\;C_{b},$$
the computational problem of compartmental analysis is, therefore, the one to determine, at each time point,
(90)$$\left(k^{*},V_{b}^{*}\right)=\arg\min_{k,V_{b}}∥C_{T}-F\left(k,V_{b}\right)∥\;.$$

In this optimization equation, ∥·∥ denotes the topology with which the distance between the experimental and predicted total concentrations is measured. Naive approaches to the computational solution of Equation (Equation 90) are typically characterized by three main drawbacks:They typically suffer numerical instabilities related to the non-uniqueness and sensitivity limitations discussed in Section 5.Since the operator F is clearly non linear and, further, the space where possible minimizers can be searched for is typically big, they may suffer local minima.Particularly, in the case of three-compartment models, the number of kinetic parameters to determine is high, which implies that they are computationally demanding.

Several numerical methods have been applied for the solution of Equation (Equation 90), whose reliability and computational effectiveness depend on the choice of the topology ∥·∥ and by the way possible prior information on the solution are encoded in the optimization process. Further, the computational algorithms utilized for solving the minimization problem typically belong to three general approaches: the deterministic, statistical, and biology-inspired ones. In the following, we will provide a sketch of the main computational aspects of these three approaches, assuming that $V_{b}$ is known thanks to either experimental or physiological information (the generalization to the case when also $V_{b}$ is an unknown parameter is straightforward).

### 7.1. Deterministic Approaches

Most deterministic approaches utilize numerical methods to solve the minimum problem [50]
(91)$$k^{*}=\arg\min_{k,V_{b}}\{∥C_{T}{-F\left(k\right)∥}_{2}+{\lambda ∥k∥}_{p}^{p}\}\;,$$
where λ is the so-called regularization parameter tuning the trade-off between the fitting capability of the algorithm and its numerical stability. The least-squares problem corresponding to λ=0 is often addressed either by means of the standard Levenberg-Marquardt scheme [51] or by using generalized separable parameter space techniques [52,53]. Other deterministic methods re-formulate the compartmental problem as the non-linear zero-finding problem
(92)$$F_{T}=0\;,$$
with
(93)$$F_{T}:=C_{T}-F\left(k\right)\;$$
and apply an iterative Gauss-Newton approach for its solution [13,54]. Finally, more recently, a regularized affine-scaling trust-region optimization method has been introduced to solve the compartmental method in a rapid fashion, so that applications to parametric imaging are possible and computationally effective [55].

### 7.2. Statistical Approaches

The general framework where statistical approaches are formulated is the Bayes theorem, which, in this context, can be written as
(94)$$\pi \left(k|C_{T}\right)=\frac{\pi \left(C_{T}|k\right)\pi \left(k\right)}{\pi (C_{T})}\;.$$

In this equation, $\pi (C_{T}|k)$ is the likelihood distribution depending on F and on the statistical properties of the noise affecting the measurements (which is Poissonian); π(k) is the prior distribution encoding all the a priori information at disposal on the solution; $\pi (C_{T})$ is a normalization factor. The posterior distribution $\pi (k|C_{T})$ is the solution of the compartmental inverse problem, which can be utilized to compute k via either the maximum a posteriori
(95)$$k_{map}=\arg\max_{k}\pi \left(k|C_{T}\right)\;,$$
or the conditional mean
(96)$$k_{CM}=\int \pi \left(k|C_{T}\right)dC_{T}\;.$$

If the prior distribution is just concerned with the positivity of the solution, it is well-established that $k_{map}$ can be determined by means of the expectation maximization iterative scheme [14,56,57]. Encoding more sophisticated prior information in the Bayesian framework requires the implementation of more sophisticated Monte Carlo schemes for the computation of the posterior distribution [58,59,60,61,62,63].

### 7.3. Biology-Inspired Approaches

Some recent approaches to the computational solution of tracer kinetics problems exploit optimization schemes that rely on biological inspiration. An example of how this perspective may be helpful in compartmental analysis is illustrated in Reference [49], where an optimization scheme inspired by ant colony behavior is utilized to determine the kinetic parameters. However, most optimization algorithms belonging to this group of methods rely on neural networks that are formulated within the framework of machine and deep learning theory [64,65,66].

In order to show how some of these methods behave in action, Figure 8 summarizes some results obtained in the literature by using experimental measurements recorded by means of a PET scanner for small animals. These results refer to the four CMs discussed in Section 6. Specifically, in Figure 8 the parameter values obtained in panel (A) refer to the reference tissue model in Figure 5 and have been obtained by means of a deterministic Gauss-Newton scheme [48]. An analogous deterministic algorithm has been utilized in panel (B) [16] to compute the parameter of the liver physiology illustrated in Figure 6. Panel (C) and Panel (D) refer to the renal physiology modelled in Figure 7 and the parameters have been computed by means the biology inspired Ant Colony Optimization algorithm [49] and by the statistical Expectation Maximization iterative scheme [14], respectively. Finally, panel (E) contains the tracer parameters provided by a regularized Gauss-Newton method [33] in the case of the model including the endoplasmic reticulum illustrated in Figure 2. Just a few comments on these results: most standard errors are rather small, which shows the stability effects related to the introduction of regularization in the optimization process; the small value of $k_{pf}$ in panel (B) confirms the fact that metformin leave this parameter essentially unaltered (indeed, these results have been obtained in the case of six tumor models treated with metformin); panel (C) and panel (D) show the effect of metformin on $k_{bf}$; panel (E) shows that the analogous parameter for the model accounting for the role of the endoplasmic reticulum (i.e., $k_{2}$) significantly decreases with respect to the case when the reticulum is neglected.

Finally, we remark that a Matlab package equipped with a user-friendly graphical user interface (GUI) for the analysis of different compartmental models, namely standard 2-CM, kidney, and liver, is available at https://github.com/theMIDAgroup/CompartmentalAnalysis. Similarly, a Matlab package for analyzing a compartmental system based on reference tissue modeling is available at https://github.com/theMIDAgroup/ReferenceTissue_CompartmentalAnalysis, while the github repository https://github.com/theMIDAgroup/BCM_CompartmentalAnalysis contains the Matlab code for the analysis of the 3-CM accounting for the endoplasmic reticulum. The last accessed date to all the aforementioned websites is the 5th of August 2021.

## 8. Conclusions

Compartmental analysis is a well-established approach to the interpretation of dynamical FDG-PET data and this review paper has aimed to point out the fact that numerical algorithms for the reduction of compartmental models play a crucial role for the comprehension of cancer glucose metabolism from a quantitative viewpoint. Yet, some technical issues are still open, whose solution would imply a further significant improvement in the comprehension of glucose dynamics in cancerous tissues.

First, the approaches illustrated in this review all rely on an indirect perspective, in which the compartmental analysis is performed on the reconstructed PET images. However, direct parametric imaging [58,67] can be realized by using the raw PET sinograms as input and then by solving the inverse problem relating the kinetic parameters to such sinograms. These one-shot approaches are not currently employed for the systematic analysis of FDG-PET data, mainly because they would have to account for the intertwining of temporal and spatial correlations, which might increase the complexity of the optimization process. However, they certainly present significant potential advantages, since they may exploit the use of input data characterized by a well-established statistical distribution and a higher signal-to-noise ratio.

Second, image processing could improve the numerical solution of the compartmental equations by introducing morphological information that can be exploited in the computation of the total concentration. More specifically, a possible scheme would rely on a segmentation step applied on the co-registered CT image that automatically identifies the voxels corresponding to the organ (namely, the tumor) and by the generation of a binary map that is multiplied against the FDG-PET reconstructed map in order to allow an accurate computation of the total concentration.

Finally, all methods described in this review explicitly assume that the tracer concentration in blood at the beginning of the compartmental experiment is zero. This is reflected into a vanishing initial condition in the Cauchy problem that significantly simplifies the computation. However, in clinical applications the initial concentration is not zero and a significant improvement of the reliability of the compartmental models would be gained by dealing with a non-vanishing Cauchy condition as a further unknown in the differential problem. 

## Figures and Tables

**Figure 1 metabolites-11-00519-f001:**
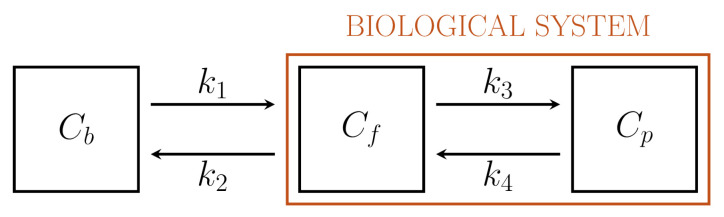
Standard Sokoloff’s 2-compartment model.

**Figure 2 metabolites-11-00519-f002:**
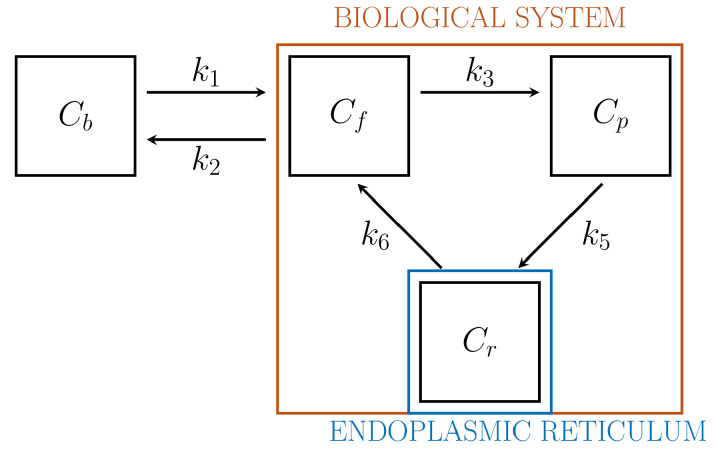
3-compartment model for cell absorption.

**Figure 3 metabolites-11-00519-f003:**
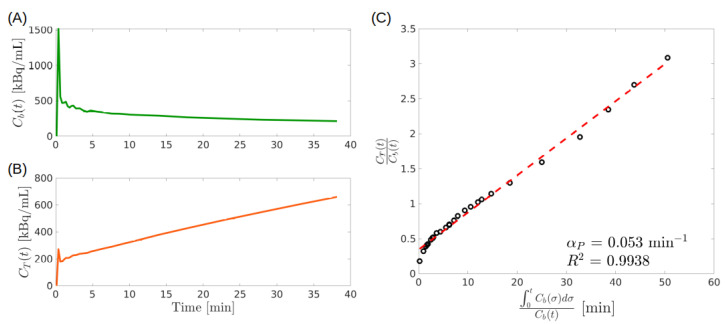
Illustrative example of the Patlak graphical approach. (**A**) Experimental input function $C_{b}\left(t\right)$ as a function of time. (**B**) Simulated total concentration $C_{T}\left(t\right)$ as a function of time. (**C**) Patlack plot (black circles). The red dotted line is the results of a linear regression model fitted to the points of the plot. The value of the corresponding slope $\alpha_{P}$ and of the coefficient of determination $R^{2}$ are also reported.

**Figure 4 metabolites-11-00519-f004:**
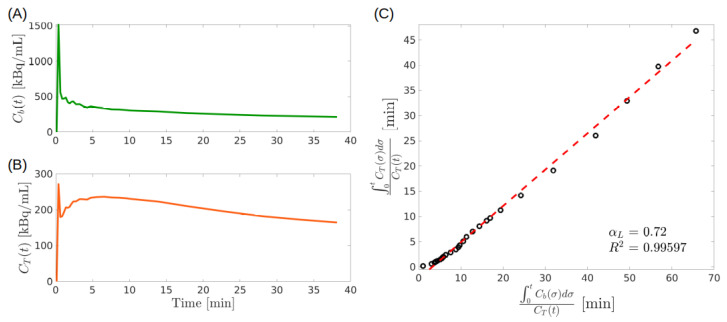
Illustrative example of the Logan graphical approach. (**A**) Experimental input function $C_{b}\left(t\right)$ as a function of time. (**B**) Simulated total concentration $C_{T}\left(t\right)$ as a function of time. (**C**) Logan plot (black circles). The red dotted line is the results of a linear regression model fitted to the points of the plot. The value of the corresponding slope $\alpha_{L}$ and of the coefficient of determination $R^{2}$ are also reported.

**Figure 5 metabolites-11-00519-f005:**
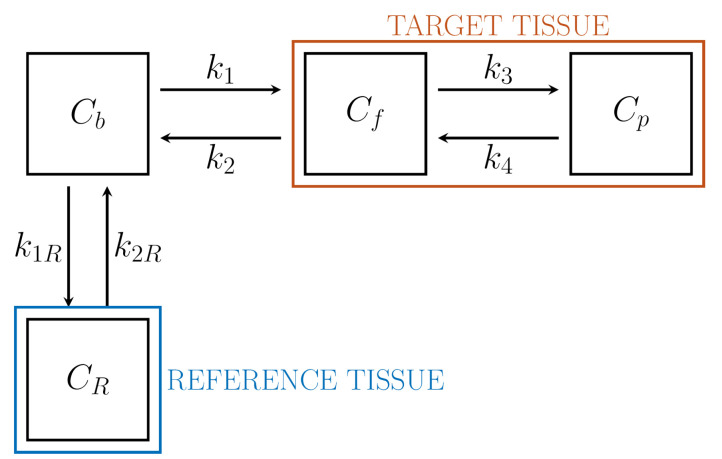
Compartment model for the reference tissue.

**Figure 6 metabolites-11-00519-f006:**
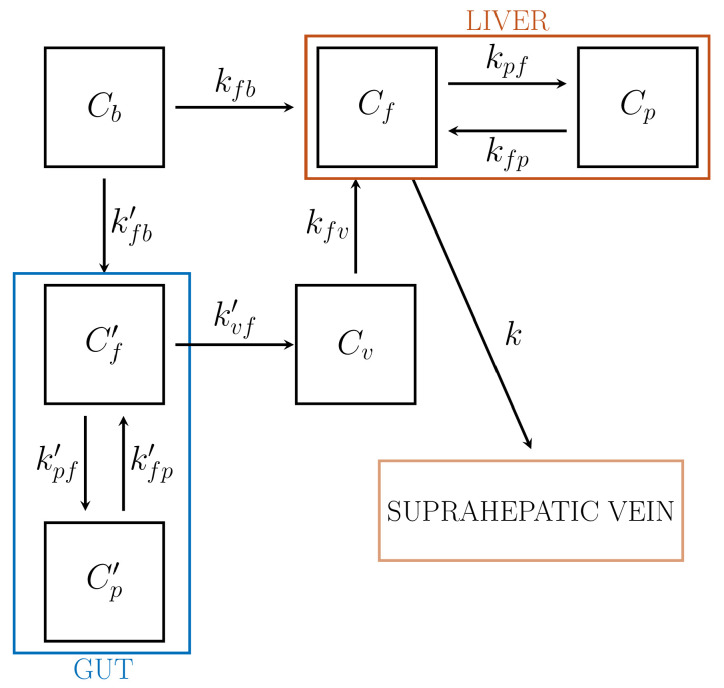
Compartment model for liver.

**Figure 7 metabolites-11-00519-f007:**
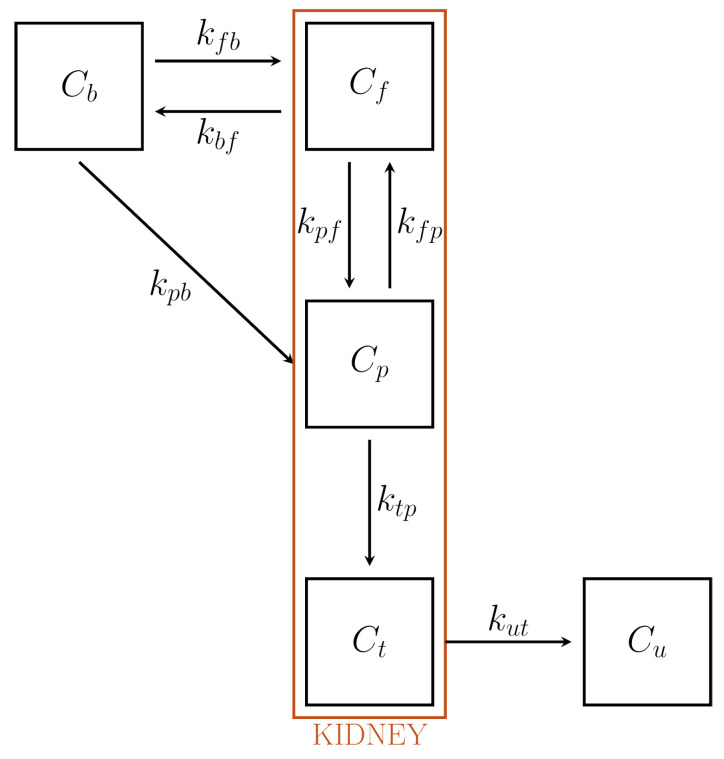
Compartment model for the renal system.

**Figure 8 metabolites-11-00519-f008:**
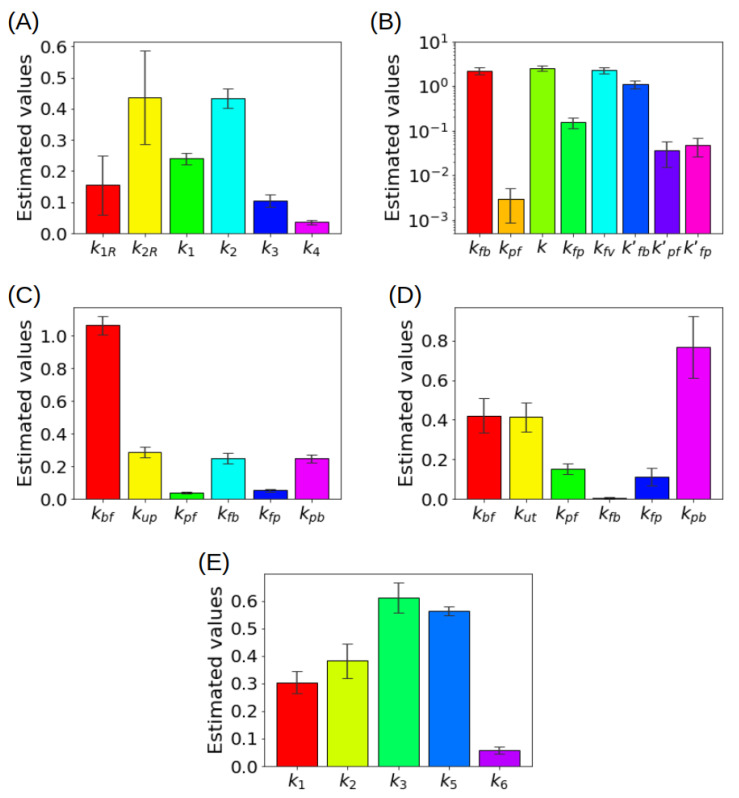
(**A**) Kinetic parameters estimated for the reference tissue CM with a deterministic approach (**B**) Kinetic parameters estimated for the liver CM with a deterministic approach (**C**) Kinetic parameters for a simplified CM for the renal system estimated with ant colony optimization, (**D**) Kinetic parameters for the CM of the renal system estimated with a statistical approach (**E**) Kinetic parameters of the CM including the endoplasmic reticulum estimated with a regularized Gauss-Newton approach.

**Table 1 metabolites-11-00519-t001:** Comparison among different compartment models. For each of the models we summarize the main characteristic (*Scope*), the involved tracer compartments (*Compartments*), and the number of unknow rate constants to be estimated (*Size of* k). The last to columns indicate whether the Patlak graphical approach can be applied (*PGA*), and whether the identifiability of the considered model has been proved (*Identifiability*).

Model	Scope	Compartments	Size of *k*	PGA	Identifiability
2-CM	Basic standard model	$C_{f}$, $C_{p}$	4	✓	✓
3-CM	Focus on endoplasmic reticulum	$C_{f}$, $C_{p}$, $C_{r}$	5	✓	✓
RTM	Avoids use of IF	$C_{f}$, $C_{p}$, $C_{R}$	6	✕	✓
Liver	Role of gut for the dual input (HA and PV)	$C_{f}$, $C_{p}$, $C_{f}^{\prime }$, $C_{p}^{\prime }$, $C_{v}$	8	✕	✕
Kidney	Focus on tubules and bladder	$C_{f}$, $C_{p}$, $C_{t}$, $C_{u}$	7	✕	✓

CM = Compartment Model; RTM = Reference Tissue Model; IF = Input Function; HA = Hepatic Artery; PV = Portal Vein.

## Data Availability

The data presented in this study are available in article.

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
