# Peer review of "Mathematical Models for FDG Kinetics in Cancer: A Review"

_metabolites, 2021, doi:10.3390/metabo11080519_

Round 1
Reviewer 1 Report
Although this paper has compiled many research models, it lacks comparisons of dynamical data of FDG-PET/the comprehension of cancer glucose metabolism or other indicators. It does not show the comparative advantages and disadvantages of various models.
Other minor suggestions:
1. Figure 5 needs to add a Y-axis label.
2. Some typos need to be checked, such as p.25 line 699 "TThese results..."
Reviewer 2 Report
The authors present great overview on mathematical modeling on compartmental modeling. Starting from simple models to more advanced, with nice explanations. I especially appreciate inclusion of the optimization methods and graphical models. Despite the context is very rich on information, it is well explained and easy to follow. I sort of feel like I just undergo crushed course on compartmental modeling, and I believe other readers will benefit from this great overview.
I have few minor comments.
- In the Patlak and Logan graphical approach. I think the methods are explained well, but being it a graphical approach it would be great if authors could include visual example (or point the reader to such resources).
- It might be useful to the reader if authors could point us to some computational libraries and packages that could be used for compartmental analysis (if available of course).
Typos:
- Page 1, line 31: typo: "densito"
- Page 4: line 146: application OF a mathematical models
- Page 5: line 218: diffusion may BE described by application of a continuous model.
- Page 8: line 314: (2-CM) should be 3-CM
Author Response
please see attchment

Round 2
Reviewer 1 Report
Thank you for addressing my previous questions.